



# FastLE:A New Load Extrapolation Method for Site-specific Wind Turbines Using the Load Distribution Meta Model

Pengfei Zhang [1], Shanshan Guo [1], Zuoxia Xing [1,*], Qingqi Zhao [2], and Qiang Jin [3]

[1]College of Electrical Engineering, Shenyang University of Technology, Shenyang 110870, China
[2]Skills Training Center, State Grid Liaoning Electric Power Co. Ltd., Jinzhou 121001, China
[3]Huaneng Clean Energy Research Institute, Beijing 102209, China

*Correspondence to*: Zuoxia Xing (xingzuox@163.com)

**Abstract.** To ensure the safety of wind turbines at specific sites, IEC 61400-1 mandates the extrapolation of loads as a key
requirement. Given the variability in wind parameters across different turbine sites, particularly in complex terrains, this task demands significant computational resources for simulations. However, the method recommended in the standard fall short of providing comprehensive assessments and rapid iterations necessary for all turbine locations within wind farm optimization designs. This paper presents a rapid load extrapolation method, named FastLE, which is based on a load distribution meta-model and tailored for specific sites. Based on 20 test cases, the blade root out-of-plane bending moment
(OOPBM) for a 50-year return period was calculated using both the IEC method and the FastLE method introduced in this paper. Through comparative analysis, the mean APE is only 3.165 %, and the computation time for a single calculation has been reduced from 20 hours to less than 1 second. The results show that the FastLE method can complete load extrapolation calculations for wind turbines in seconds with high accuracy. This makes it suitable for ensuring structural integrity during iterations of wind farm layout optimization or turbine type optimization, thereby reducing the safety risks associated with
wind turbines.

## 1 Introduction

With the rapid development of wind energy, a growing number of new wind farms is being designed and constructed worldwide(World energy outlook 2024). However, the frequency of safety incidents caused by load-related issues is also rising(IEC 61400-1, 2019; Paul et al., 2019). As a result, the crucial role of load extrapolation in ensuring the structural
integrity of wind turbines has garnered significant attention(Cao et al., 2018). For wind turbines, the loading conditions are contingent upon the turbulent inflow of wind across a spectrum of atmospheric conditions. Consequently, statistical extrapolation is essential for projecting long-term load profiles from limited simulation datasets. This predictive exercise is crucial for determining the load rates associated with key design scenarios outlined by the International Electrotechnical Commission (IEC) standards for wind turbine design(IEC 61400-1, 2019). Similarly, it is common for different sites within
the same wind farm to experience varying wind conditions. Therefore, it is essential to calculate the extrapolated loads for





each individual site and compare them with the design loads to ensure the safety of the wind turbines(IEC 61400-1, 2019). The IEC-recommended approach requires a minimum of 60 seeds to simulate 600 s load time series under normal operating conditions with high wind speeds. Moreover, it demands significant computational resources and considerable time for performing a 50-year extrapolation based on the distribution of load extremes(Fogle et al., 2008). This intensive requirement

complicates the consideration of extrapolated loads during the iterative optimization of wind farm layouts and turbine types, leading to suboptimal solutions(Zhang et al., 2024; He et al., 2024) or protracted optimization periods(Sarcos et al., 2024). To address this challenge, there is a pressing need for a fast method to calculate extrapolated loads tailored to specific site wind conditions. This method would be essential for assessing safety risks associated with loads at various sites within a wind farm during the design phase. Furthermore, it would enable iterative optimizations in which extreme extrapolated loads

serve as crucial constraints for enhancements in wind farm layout and turbine types.

Numerous studies have investigated methods for calculating extreme extrapolated loads. The latest version of the IEC 61400-1 standard mandates the calculation of these loads as a design requirement and presents two computational pathways: "fitting before aggregation" and "aggregation before fitting"(IEC 61400-1, 2019). Both approaches require extensive load simulation data under normal operating conditions across various wind speeds, making them widely adopted in the industry.

Toft and Sørensen et al.(2011) has compared these two approaches and concluded that "fitting before aggregation" yields superior results, making it highly applicable for assessing load safety at specific sites. Saranyasoontorn et al.(2006) introduced the Environmental Contour Method (EC) for coupling wind speed distributions with extreme load distributions, achieving promising results. Nataraja et al.(2008) utilized Quadratic Distortions to reduce the uncertainty in extrapolated loads. The IEC 61400-1 standard recommends using the IFORM method(IEC 61400-1, 2019). To validate the applicability

of various methods, Moriarty(2008) from NREL established two comprehensive load simulation databases that cover a wide range of wind parameters. These databases were used in the IEC Loads Extrapolation Evaluation Exercise. The studies in Toft and Naess et al.(2011) and Schinas et al.(2021) further reduce the uncertainty in calculating extrapolated loads. However, there is limited research on improving computational speed. On the other hand, in the IEA Wind Task 37(Dykes et al., 2017) on wind turbine and wind farm systems engineering, load constraints are essential during the optimization process.

Nikolay et al. (2018) from DTU conducted an in-depth study on rapid assessment methods for wind turbine loads and proposed a surrogate model called Wind2Load, based on Kriging regression model and Polynomial Chaos Expansions (PCE), for load assessment. This approach significantly improves computational speed compared to traditional aero-elastic simulation tools and can be used for load safety calculations at various locations within a wind farm. However, it does not address extrapolated loads. Duthé et al.(2024) employed Graph Neural Networks (GNN) and transfer learning to establish an

effective fatigue load assessment model for wind farms. Similarly, Singh et al.(2024), Bossanyi(2022), Guilloré et al.(2024) and Pettas et al.(2024) have utilized machine learning techniques to accelerate load assessment processes. In summary, there is a notable gap in the rapid calculation methods for site-specific extreme extrapolated loads, and the application of machine learning presents an effective solution to this challenge.





Therefore, this paper proposes a method for calculating extrapolated loads of wind turbines at specific sites based on load distribution meta models. This method can effectively fill the gap mentioned above and can be used for rapid implementation of load safety constraints in wind farm layout covering all turbine sites and during optimization iterations. The study aims at fulfilling 6 points as following: 1, The load distribution meta-model database created using load time series is obtained under normal operating conditions. The Monte Carlo sampling method is employed to ensure a broad representation of wind parameters while working with a limited number of samples. 2, Extraction and validation of independent load sample. 3, Optimal selection of local load distribution. 4, Post-processing of extrapolated extreme loads, the mutual progression among local load distribution, extreme load distribution, and long-term load distribution. 5, Training and optimization of the load distribution meta-model. And test cases.

This study employs the out-of-plane bending moment (OOPBM) at the blade root under normal operating conditions for a 50-year extreme load extrapolation, which includes Design Load Cases (DLCs) 1.1 as specified in IEC 61400-1:2019. The research focuses on the WTG156-4.55 wind turbine model, with loads generated through simulations using Bladed software.

## 2 Load Database for Extrapolation

The data utilized for extrapolation methods is derived from time series simulations of the turbine operating across a specified wind range. For specific wind parameters, simulations are conducted according to IEC standards for DLC 1.1 load cases, with a minimum duration of 10 minutes per time series. The resolution of wind speed is set at 2 m s-1 intervals from cut-in to cut-out wind speed. A minimum of 15 simulations is required for each wind speed interval from ($Vr$-2 m s$^{-1}$) to cut-out (generally, 60 simulations are used), and six simulations are needed for each wind speed below ($Vr$-2 m s$^{-1}$). For WTG 156-4.55, with a cut-in wind speed of 2.5 m s$^{-1}$ and a cut-out wind speed of 24 m s$^{-1}$, at least 558 simulations are required.

Aside from wind speed, the primary wind parameters affecting the loads on wind turbines include air density, turbulence intensity, wind shear, and inflow angle(Moriarty, 2008; Dimitrov et al., 2018; Kelly et al., 2014; Dimitrov et al., 2017). For the ranges and distributions of these variables, this study utilizes data from 541 meteorological towers in a specific region, with the lower boundary set at the 1st percentile and the upper boundary at the 99th percentile, as shown in Fig.1.





**Figure 1: Scatter plots, density contour lines, and kernel density estimations for various wind condition parameters at a specific site. (+ indicates the measured values.)**

Regarding turbulence intensity, it varies with different wind speeds. The Eq. (1) is used to calculate the turbulence intensity for each wind speed based on $I_{ref}$.

$$I_{V_i} = \frac{I_{ref}(0.75V_i + 5.6)}{V_i}, \qquad (1)$$

According to the distribution and range of various wind parameters in Table 1, Monte Carlo sampling method was used for data collection, with a sample size of 100. As shown in Fig.2, the statistical characteristics of the sampled data are basically
consistent with those of the original data.



**Table 1 Bounds of variation for the wind parameters considered. All values are defined as annual statistics.**

| Variable | Distribution | Lower bounds | Upper bounds | unit |
|---|---|---|---|---|
| Air density | gaussian kernel density(Bashtannyk et al., 2001) | 0.992 | 1.247 | kg/m$^3$ |
| Shear | log-normal density | 0.006 | 0.422 | - |
| Inflow angle | log-normal density | -0.78 | 13.464 | degree |
| Iref | uniform density(Dimitrov et al., 2017) | 0.09 | 0.225 | - |

**Figure 2: Scatter plots, density contour lines, and kernel density estimations for various wind condition parameters (blue dot indicates measured values, orange dot indicates MC sampled values).**






The configuration of the wind turbine used in the simulation is shown in Table 2.

**Table 2 The configuration of the wind turbine WTG156-4.55.**

| parameters | unit | value | parameters | unit | value |
|---|---|---|---|---|---|
| Rated power | kW | 4550 | cut- in wind speed | m s$^{-1}$ | 2.5 |
| Rotor Diameter | m | 156.2 | cut-out wind speed | m s$^{-1}$ | 24 |
| Hub height | m | 110 | rated wind speed | m s$^{-1}$ | 10.8 |
| design class | - | IIIA | Rotor rated speed | rmp | 9.5 |
| power regulation method | - | pitchable and variable speedta | Rotor speed range | rmp | 6-9.5 |

Using the Bladed software, a simulation was conducted for $558 \times 100$ normal operating conditions over a duration of 10 minutes each, generating a database for load extrapolation. In accordance with wind turbine design standards, the analysis of

load extrapolation concerning the structural integrity must include at least the computation of extreme values for the blade root in-plane bending moment, out-of-plane bending moment, and tip deflection, as shown in Fig.3. This paper focuses on the methodological exposition, so the out-of-plane bending moment at the blade root will be analyzed as an example.

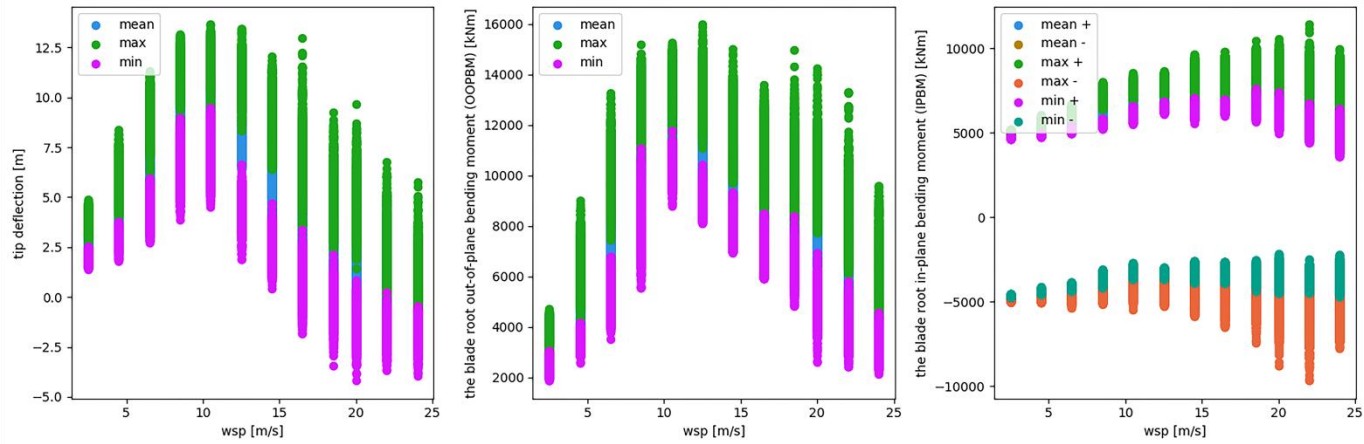

**Figure 3: Basic statistical characteristic parameters under different wind speeds for sample: Tip Deflection (left), Out-of-Plane**
**Bending Moment (OOPBM) (middle), and In-Plane Bending Moment (IPBM) (right).**

## 3 IEC standards recommend load extrapolation method

In accordance with IEC 61400-1:2019 and relevant literature, Figure 4 illustrates the load extrapolation process adopted in this paper.



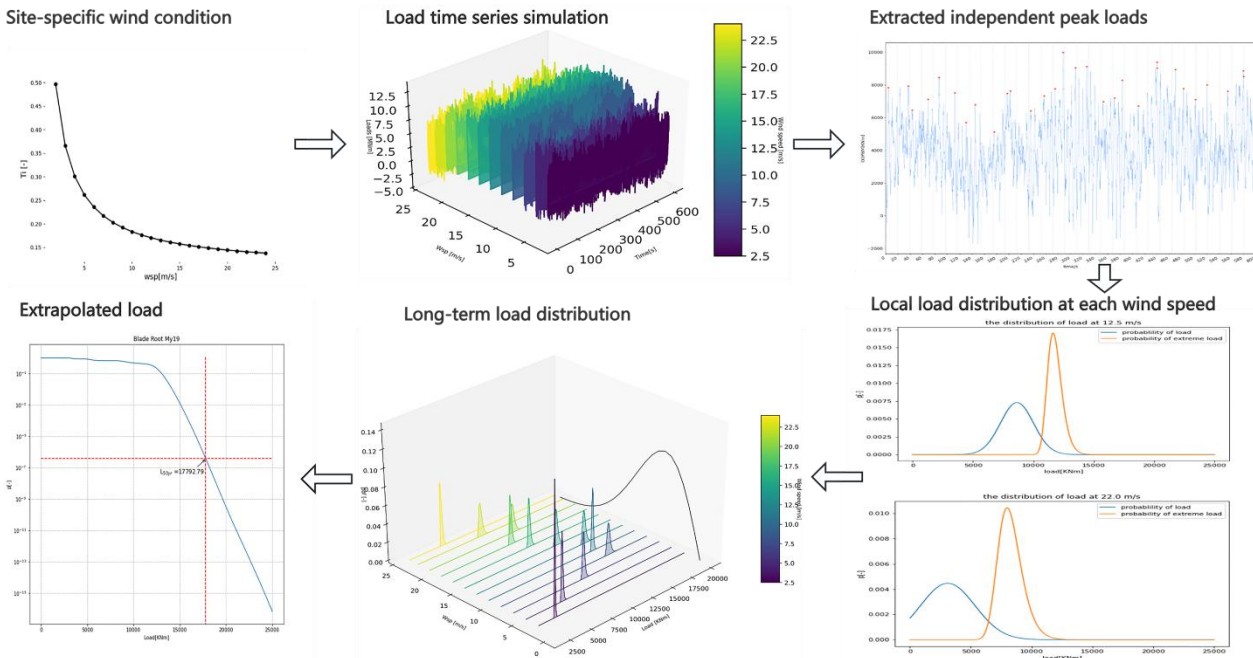

**Figure 4: The benchmark method for the load extrapolation process.**

The benchmark method utilized in this paper for load extrapolation at specific sites is outlined as follows:

1, The load time series simulation is run for different wind speeds, from cut-in to cut-out wind speeds, under normal power production conditions. This simulation is based on the site-specific wind parameters which include air density, turbulence intensity at different wind speeds, wind shear, inflow angles, etc. A total of 558 simulated time series loads, each lasting 600

s, are produced when the wind speed interval is set at 2 m s$^{-1}$.

2, The "fitting before aggregation" method can be used to ascertain the long-term distribution of the extremes. The long-term distribution is derived by weighting these short-term distributions in accordance with the wind distribution, and the local distribution is fitted to the peaks at each wind speed.

3, Peaks are extracted from the load time series using the Block Maxima method. The 600-second time series is divided into

20 blocks and each lasts 30 s. This block length is adequate to ensure the independence of the peaks, as detailed in Zhang et al.(2024).

4, The local distribution function of the extracted peak loads is fitted using the maximum likelihood method. The following is the probability density function (PDF) of the Weibull distribution.

$$f(x; \lambda, k, \gamma) = \frac{k}{\lambda}\left(\frac{x-\gamma}{\lambda}\right)^{k-1} e^{-(x-\gamma/\lambda)^k}, \quad x \geq \gamma, \tag{2}$$



Where,$\lambda$ (Scale parameter) controls the scale of the distribution. $k$ (Shape parameter) determines the shape of the distribution. $\gamma$ (Location parameter) represents the threshold below which the probability density is zero.

The extreme distribution for the maximum response $L$ throughout the time interval [0,T] is obtained from the distribution of local peaks as follows:

$$F_{extreme}(L|V) = F_{local}(L|V)^N, \tag{3}$$

where $N$ is the expected number of independent peaks at the wind speed $V$ during the time interval [0,T].

5, By integrating over the wind speeds, as described by the Rayleigh distribution, the extreme distribution can be used to derive the long-term distribution for the maximum response within the time interval [0,T].

$$F_{long-term}(L) = \int_{v_{cut-in}}^{v_{cut-out}} F_{extreme}(L|V)f(V)dV, \tag{4}$$

and

$$P(V) = 1 - exp\left[-\pi\left(\frac{V}{2V_{ave}}\right)^2\right], \tag{5}$$

where, $P(V)$ is the cumulative probability function, $f(V)$ can be calculated, $V$ is the wind speed;$V_{ave}$ is the average value of $V$.

6, The exceedance probability of the 50-year extreme load is estimated as follows.

$$L_{50year} = F_{long-term}^{-1}\left(\frac{1}{50\times365\times24\times6}\right), \tag{6}$$

**4 FastLE Method**

The standard extreme load extrapolation method was employed to simulate the load conditions for normal power production of the WTG156-4.55 turbine using Bladed 4.10.0.22. Each simulation took approximately 70 min. Utilizing a 32-core CPU as the computational resource, the total time required for the calculations would be around 20.344 h. In wind farms containing several, dozens, or even hundreds of turbines, the time-intensive nature and substantial computational resource

demands of this approach are clearly impractical. Consequently, overly conservative designs are often employed, which reduces the economic viability of projects. To address these challenges, this paper introduces a rapid extrapolation method for extreme loads, utilizing a load distribution meta-model called FastLE.

**4.1 Independence Testing**

As we all know, ensuring that block maxima chosen from each time series are independent of each other is crucial for the

statistical extrapolation method. Blum et al.(1961)'s test has been applied to wind turbine load extrapolation by Fogle et



al.(2008). It was discovered that block sizes of approximately 10-15 s for OOPBM produced independent block maxima when evaluated at the 1 % significance level using a statistical test.

To balance the large amount of data needed to fit the local load distribution with the requirement to reduce sample numbers for maintaining independence, we introduce DcorrX (Cross Distance Correlation) as an additional test for
independence(Shen et al., 2024). DcorrX is an independence test between two time series, where the population parameter equals zero if and only if the time series are independent. This method is grounded in the concept of energy distance between distributions.

Let $x$ and $y$ be $(n, p)$ and $(n, q)$ series respectively, which each contain $y$ observations of the series $X_t$ and $Y_t$. Similarly, let $x[j: n]$be the $(n - j, p)$ last $n - j$ observations of $x$. Let $y[0: (n - j)]$ be the$(n - j, p)$ first $n - j$ observations of $y$. Let $M$ be
the maximum lag hyperparameter. The cross distance correlation is defined as follows:

$$DcorrX_n(x, y) = \sum_{j=0}^{M} \frac{n-j}{n} Dcorr_n(x[j: n], y[0: (n - j)]),$$ (7)

$$Dcorr_n(x, y) = \frac{Dcov_n(x,y)}{\sqrt{Dcov_n(x,x) \cdot Dcov_n(y,y)}},$$ (8)

$$Dcov_n(x, y) = \frac{1}{n(n-3)} tr(C^x C^y),$$ (9)

$$C_{ij}^x = \left[ D_{ij}^x - \frac{1}{n-2} \sum_{t=1}^{n} D_{it}^x - \frac{1}{n-2} \sum_{s=1}^{n} D_{sj}^x + \frac{1}{(n-1)(n-2)} \sum_{s,t=1}^{n} D_{st}^x \right] \mathbb{1}_{i \neq j},$$ (10)

$D_{ij}^x$ and $D_{ij}^y$ represent the distance between the *i-th* and *j-th* observations in samples $x$ and $y$.

For a total of 55800 samples collected under normal power production conditions, the data were categorized into three types based on wind speed: the full wind speed range, low wind speed ($<=10$ m s$^{-1}$), and high wind speed ($>10$ m s$^{-1}$). The independence of these three sample groups was assessed using DcorrX. As illustrated in the Fig.5, with a block size of 30 seconds, 74.65 % of the overall samples were found to be independent, 61.1 % of the low wind speed samples were
independent, and 84.33 % of the high wind speed samples were independent. It is widely recognized that lower wind speeds contribute minimally to the tails of long-term load distributions. Consequently, increasing the block size to guarantee independence at these low wind speeds does not aid in achieving our ultimate objective of statistical load extrapolation. Based on the test results, the final block size was selected to be 30 seconds, which ensures that the majority of load peaks are independent.






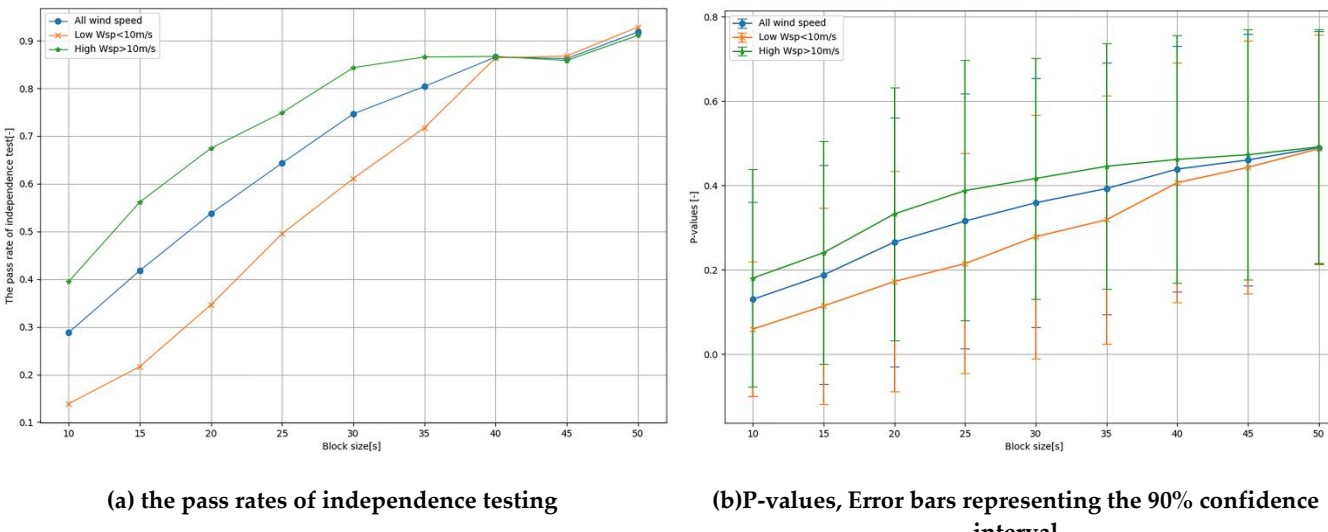

| (a) the pass rates of independence testing | (b)P-values, Error bars representing the 90% confidence interval |

**Figure 5: the pass rates and P-values for the DcorrX test under different block sizes.**

**4.2 Local Load Distribution Fitting model selection**

The local distribution function of extracted peak loads is typically modeled using a Weibull distribution(Fogle et al., 2008; Yang et al., 2022). However, Normal, Log-normal, and Gumbel distributions are also employed in fitting load distributions.

Due to the varying dynamic effects of wind turbines, the distribution of peak loads can vary at different wind speeds, as illustrated in the Fig.6.

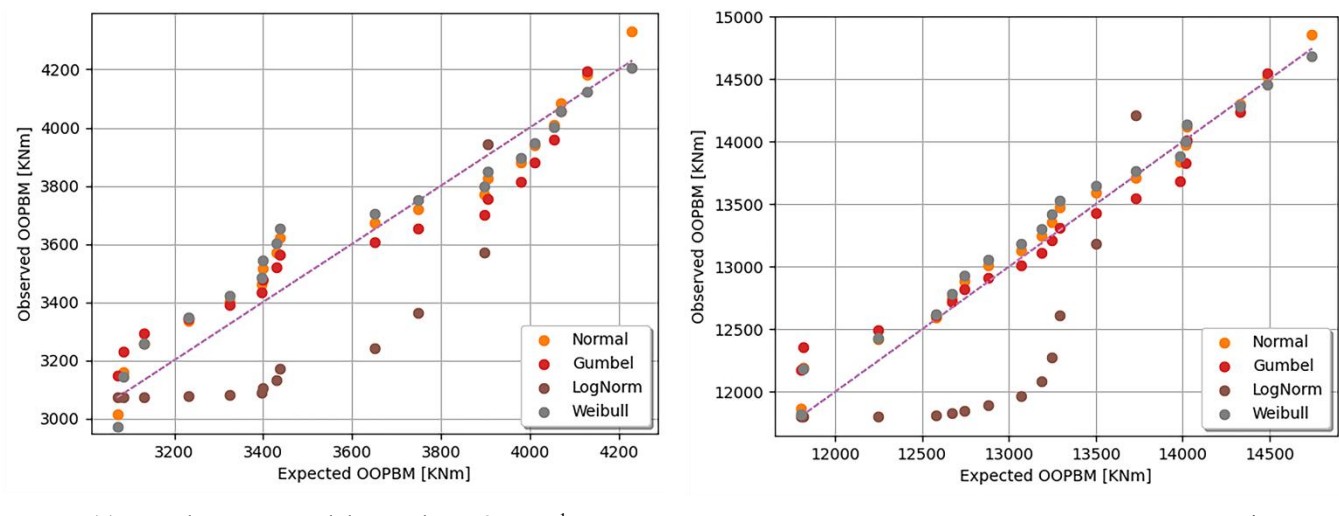

| (a) Sample versus model QQ-plot at 2.5m s⁻¹ | (b) Sample versus model QQ-plot at 10.5m s⁻¹ |





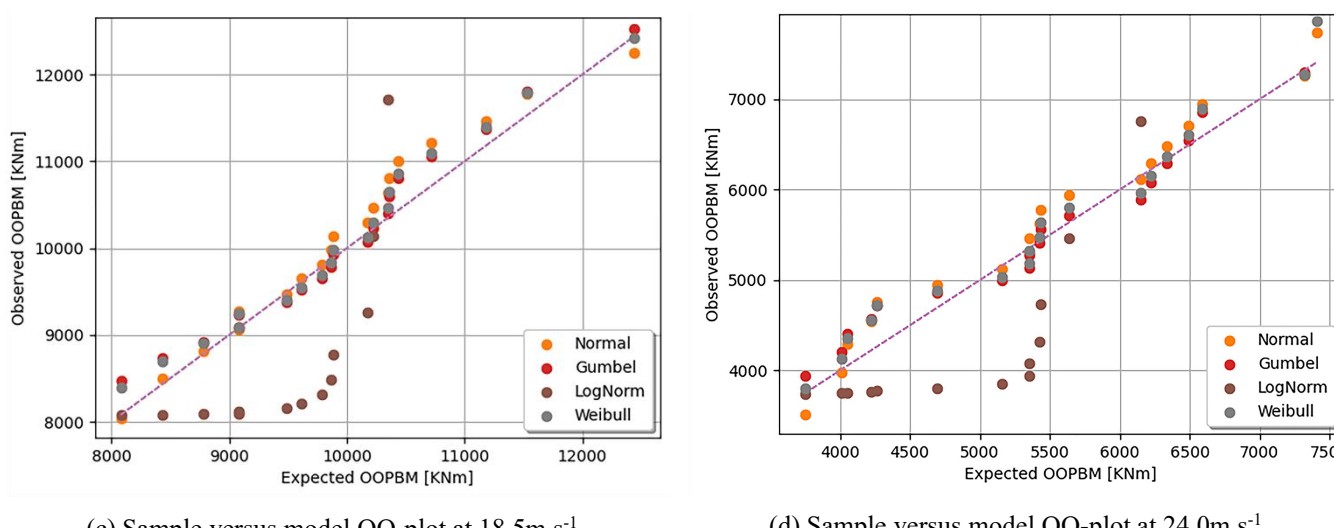

(c) Sample versus model QQ-plot at 18.5m s⁻¹     (d) Sample versus model QQ-plot at 24.0m s⁻¹

**Figure 6: QQ-plots(Quantile-Quantile plots) of the peak loads under different wind speeds and different distributions.**

The quality of the load distribution fitting is critical for the extrapolation of extreme loads. In this study, the extracted peak loads from 55800 normal power production conditions, with a block size of 30 s, are fitted using the maximum likelihood method to Weibull, normal, Gumbel, and log-normal distributions to determine their respective parameters. A Chi-squared test(Burnham et al., 2002) is then performed to validate these fits. The optimal model is selected based on the Kolmogorov-Smirnov(Dixon et al., 1983) goodness-of-fit test. The results are detailed in the Table 3 below.

**Table 3 the result of Chi-Squared test and the Kolmogorov-Smirnov goodness-of-fit test for different distributions.**

| The pass rate of 55800 samples | | | All samples | low wind speed (<= 10 m s⁻¹) | high wind speed (>10 m s⁻¹) |
|---|---|---|---|---|---|
| Chi-Squared test | Weibull | TRUE | 99.958 % | 99.900 % | 100.000 % |
| | | p-value(1 %level) | 0.673880844 | 0.657031685 | 0.685915958 |
| | Normal | TRUE | 99.903 % | 99.767 % | 100.000 % |
| | | p-value(1 %level) | 0.652822862 | 0.633291315 | 0.666773966 |
| | Gumbel | TRUE | 99.944 % | 99.867 % | 100.000 % |
| | | p-value(1 %level) | 0.582367896 | 0.523154378 | 0.624663266 |
| | Log-Normal | TRUE | 0.028 % | 0.067 % | 0.000 % |
| | | p-value(1 %level) | $1.21562\times10^{-5}$ | $2.86563\times10^{-5}$ | $3.70334\times10^{-7}$ |
| Kolmogorov-Smirnov goodness-of-fit test | Weibull | | 51.986 % | 65.667 % | 45.146 % |
| | Normal | | 33.833 % | 24.625 % | 38.438 % |
| | Gumbel | | 14.181 % | 9.708 % | 16.417 % |
| | Log-Normal | | 0.000 % | 0.000 % | 0.000 % |



### 4.3 Local Load Distribution Meta model

#### 4.3.1 Issue statement

In the process of extreme load extrapolation, simulating load time series for wind turbine normal operating conditions is the most computationally intensive and time-consuming step. To effectively speed up the load extrapolation, this study references certain literature to introduce wind parameters into the Meta model for load components. The introduction of Meta model for load extrapolation can be implemented in the following three ways, as illustrated in the Fig.7.

For Option A, this paper limits the overall training samples to just 100 due to the significant computational resources required. The most critical drawback is that the turbulence intensity at various wind speeds for specific sites does not fully align with the model defined in the IEC standards (Eq.(1)). This discrepancy necessitates including turbulence intensities at different wind speeds as inputs, which substantially increases the model's input dimensions. To adequately cover a sufficient range of wind parameters, the total number of samples would need to be expanded, thereby presenting a limitation to this option. For Option B, the training dataset comprises a total of 55800 sub-condition samples. While this is suitable for the meta model overall, it leads to an increase in the output dimensions of the meta model. With a block size of 30 seconds, the output variables for a single wind speed amount to 20 dimensions (for a 600-second load time series). Furthermore, the order of these dimensions does not affect the results, resulting in convergence issues during model training. Given the limitations in sample size and input/output dimensions, this paper chooses Option C. The training samples employ wind parameters from sub-conditions, and the output comprises the parameters of the Weibull distribution corresponding to each wind speed.

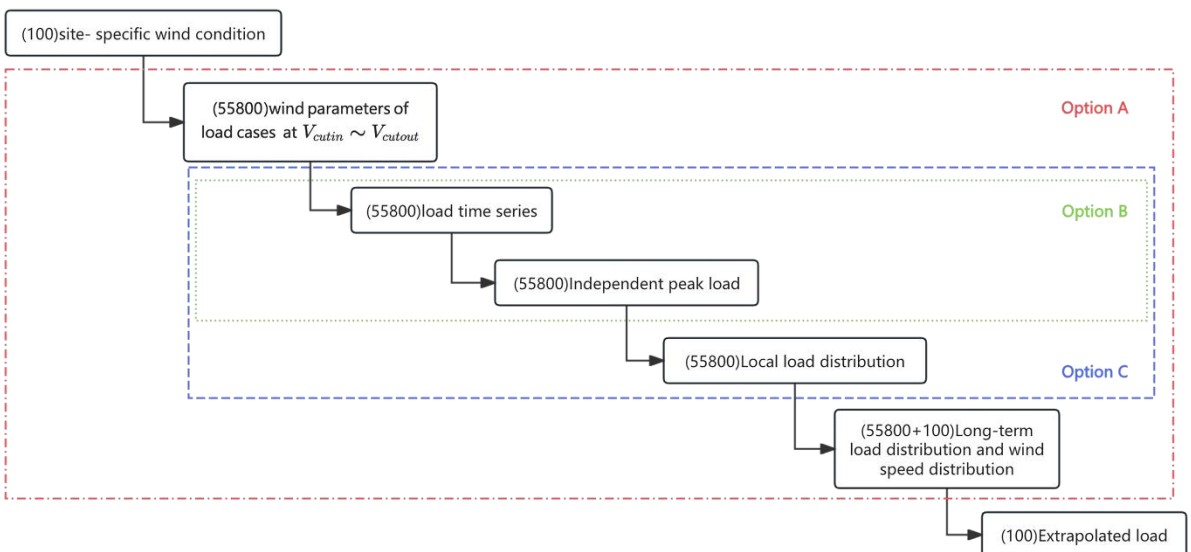

**Figure 7: Schematic diagram of the three ways to implement a meta model for load extrapolation.**





### 4.3.2 Multi-layer Perceptron

In this paper, we employ a Multi-layer Perceptron (MLP) regression model to construct the meta model. The MLP is a supervised learning algorithm designed to learn a mapping function $f: R^m \rightarrow R^o$ from a dataset, where $m$ represents the number of input dimensions and $o$ represents the number of output dimensions. Given a feature set $X = x_1, x_2, \ldots, x_m$ and a target variable $y$, the MLP can model complex, non-linear relationships for tasks such as regression. However, the MLP has certain disadvantages, which include the fact that MLPs with hidden layers have a non-convex loss function, leading to multiple local minima. This means that different random weight initializations can result in varying validation accuracies.

Additionally, MLPs require the tuning of several hyperparameters, such as the number of hidden neurons, the number of layers, and the number of iterations.

### 4.3.3 Model Training with the Optuna Optimization Framework

Due to the need for hyperparameter tuning during MLP model training, several variables are essential to improving the model's regression performance. These include the number of hidden layers, the size of each hidden layer, the activation

function used for the hidden layers, the solver for weight optimization, and the strength of the L2 regularization term, among others. To enable effective hyperparameter tuning and optimization, this paper introduces Optuna(Akiba et al., 2019), an open-source framework that automates the search for optimal parameters. The workflow is illustrated in the Fig.8 below.

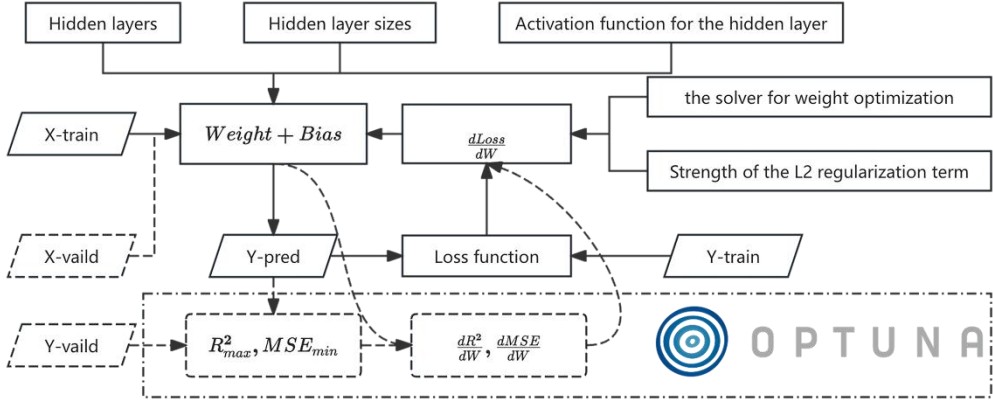

**Figure 8: The workflow of Model Training with the Optuna Optimization Framework.**

We employed 1440 validation samples and utilized Optuna for the optimization of two objectives: minimizing Mean Squared Error (MSE) (objective 0) and maximizing $R^2$ (objective 1). After 1000 iterations, the trends of both objective functions demonstrated consistent behavior, as shown in the Fig.9.


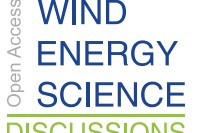


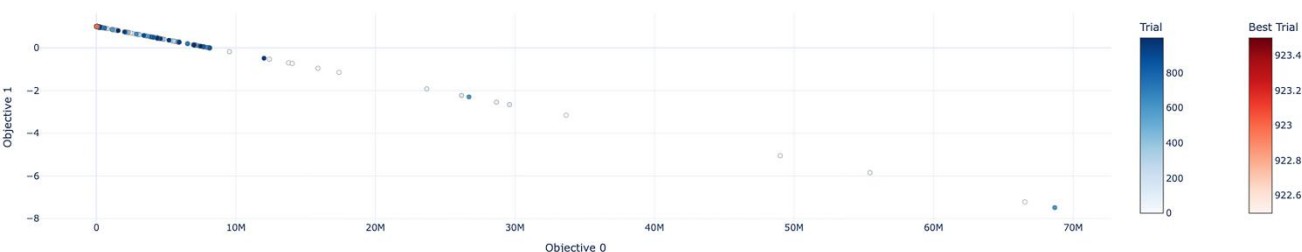

**Figure 9: Pareto-front plot for the training of a dual-objective model. (Objective 0: MSE and Objective 1: $R^2$)**

After optimization, the final optimal MLP regression model was configured with three hidden layers comprising 42, 188, and 125 neurons, respectively. The ReLU activation function was used, and the solver chosen was L-BFGS. A detailed analysis of the significance of these hyperparameters in the MLP regression task is presented in the Fig.10, which highlights the activation function as the most critical factor.

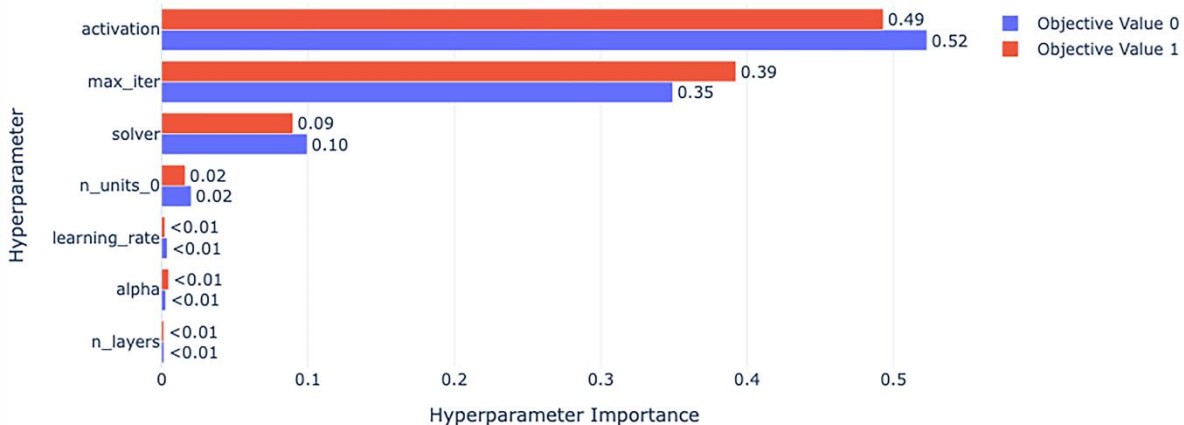

**Figure 10: Analysis of hyperparameter significance for local load distribution meta model. (Objective 0: MSE and Objective 1: $R^2$)**

The performance of the optimal model on the validation sample is illustrated in the Fig.11. As there is no significant offset in the data, the location parameter gamma is set to 0 for the Weibull distribution. For the shape parameter (denoted as w_para0 in the figure), the $R^2$ between the true and predicted values is 0.885, with a 90 % confidence interval for the error ranging from -0.22 to 0.23. For the scale parameter (denoted as w_para1 in the figure), the $R^2$ is 0.998, with a 90 % confidence

interval for the error ranging from -0.028 to 0.028.



**Figure 11: The performance of the optimal model on the validation sample.**

## 4.4 Post-processing

As shown in Fig.12, for a site-specific wind conditions, a detailed description of the interval from $V_{cutin}$ to $V_{cutout}$ with a 2 m

s$^{-1}$ interval is provided. For any given wind speed $V_i$, corresponding turbulence intensity $TI_i$, wind shear $\alpha_i$, and inflow angle $I_i$ are used as inputs for the Local Load Distribution Meta Model. This allows us to obtain the parameters for the Local Load Peaks Weibull distribution at that wind speed. Using Eq.(3), the extreme distribution can be derived. Subsequently, based on all wind speeds' extreme distributions and the Weibull probability density function, Eq.(4) is applied to determine the long-term load distribution for the site. Finally, using Eq.(5), the extrapolated load for a 50-year return period is calculated.



**Figure 12: Schematic diagram of the FastLE post-processing.**

## 5 Test case

FastLE is designed to assess the structural integrity of wind turbines during the design phase of a wind farm and does not involve measuring load data. Therefore, the test case in this paper utilizes the Monte Carlo sampling method to generate 20 sets of wind parameters, based on their respective ranges and distributions. To distinguish them from the earlier 100 training and validation datasets, the test case numbers begin at 101, as illustrated in Table 4. The turbine model used is WTG156-4.55, configured with the Bladed 4.10.0.22 software as detailed in Table 5 to simulate the OOPBM under normal operating conditions. Following this simulation, a load extrapolation was conducted to estimate conditions for a 50-year return period.



**Table 4 Wind parameter table for 20 test cases based on Monte Carlo sampling.**

| Test case | Air density [kg m$^{-3}$] | Iref | Shear | Inflow angle[°] | Test case | Air density [kg m$^{-3}$] | Iref | Shear | Inflow angle[°] |
|---|---|---|---|---|---|---|---|---|---|
| 101 | 1.0253 | 0.0930 | 0.1722 | -0.0520 | 111 | 1.0621 | 0.0923 | 0.3291 | 0.1728 |
| 102 | 1.1235 | 0.1727 | 0.1798 | -0.3124 | 112 | 1.0596 | 0.0947 | 0.2270 | 7.3205 |
| 103 | 1.0351 | 0.0902 | 0.1414 | 0.3331 | 113 | 1.0874 | 0.1148 | 0.1790 | 3.7014 |
| 104 | 1.2019 | 0.1126 | 0.3396 | 2.2621 | 114 | 1.0511 | 0.1325 | 0.1566 | 1.9260 |
| 105 | 1.1615 | 0.1200 | 0.2165 | -0.1902 | 115 | 1.0000 | 0.0942 | 0.2265 | 3.3005 |
| 106 | 1.1057 | 0.1206 | 0.1980 | 1.3117 | 116 | 1.0702 | 0.1092 | 0.2959 | 1.7365 |
| 107 | 1.0847 | 0.1199 | 0.1697 | 3.2276 | 117 | 1.0730 | 0.1330 | 0.1511 | 9.9086 |
| 108 | 1.2321 | 0.1694 | 0.1511 | 4.0153 | 118 | 1.1937 | 0.1517 | 0.1328 | 0.0902 |
| 109 | 1.0867 | 0.0916 | 0.1592 | 3.2116 | 119 | 1.2007 | 0.1046 | 0.3765 | 9.0978 |
| 110 | 1.1964 | 0.1016 | 0.2354 | 1.1041 | 120 | 1.0607 | 0.1378 | 0.2025 | 5.7453 |
| max | 1.2321 | 0.1727 | 0.3765 | 9.9086 | min | 1.0000 | 0.0902 | 0.1328 | -0.3124 |

**Table 5 Load simulation and load extrapolation configuration table.**

| Wind speed[m s$^{-1}$] | 2.5 | 4.5 | 6.5 | 8.5 | 10.5 | 13 | 15 | 16.5 | 18.5 | 20 | 22 | 24 |
|---|---|---|---|---|---|---|---|---|---|---|---|---|
| Seeds number | 6 | 6 | 6 | 60 | 60 | 60 | 60 | 60 | 60 | 60 | 60 | 60 |
| Iref | Using Iref based on Eq.(1) to calculate the turbulence intensity of esch wind speed. | | | | | | | | | | | |
| Shear | | | | | | | | | | | | |
| Inflow angle | The wind speeds are identical, each equal to the value of the wind condition. | | | | | | | | | | | |
| Air density | | | | | | | | | | | | |
| Yaw error | The values are 8, 0, and -8. each with equal probability. | | | | | | | | | | | |
| Simulation time[s] | 600 | | | | | | | | | | | |
| Load extrapolation method | Fitting before aggregation, Block size of 30 s, Local load distribution: Weibull, Wind speed distribution: Rayleigh. | | | | | | | | | | | |

Assuming all test cases employ the same wind speed probability density, modeled as a Weibull distribution with parameters A=8.463 and k=2, the results for one of the test cases are depicted in the Fig.13.





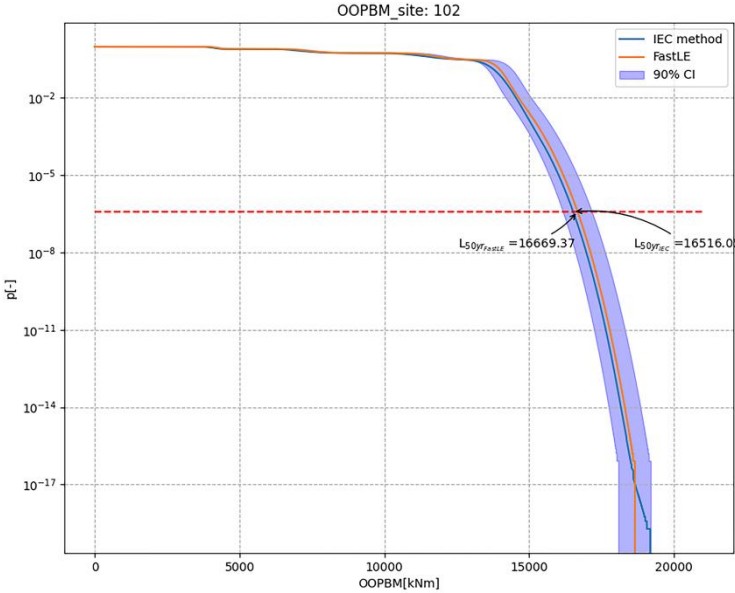


**Figure 13: Load extrapolation results for test case 102.**

For all test cases, the 50-year return period extrapolated loads for the OOPBM calculated using the FastLE method were compared to those derived from the IEC method, as illustrated in the Fig.14. The Absolute Percentage Error (APE) ranges from 0.421 % to 6.818 %, with an average of 3.165 %. When utilizing the P95 results from FastLE, the APE range narrows

to between 0.326 % and 4.288 %, with an average of 2.22 %. The entire process using the IEC standard method requires approximately 400 h solely for load simulation, whereas the complete process with FastLE takes only seconds.The results demonstrate that FastLE can maintain high computational accuracy while achieving the extrapolation of extreme loads for wind turbines in seconds. This capability can be used for structural integrity assessments of wind turbines under different wind resources and for optimizing wind farm designs with extrapolated loads as constraints.

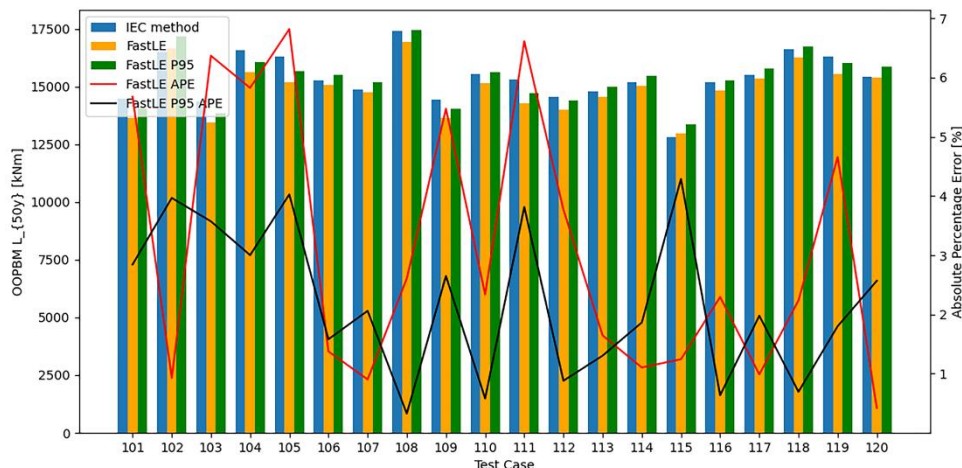


**Figure 14: the load extrapolation results using the FastLE and IEC methods for 20 test cases.**



# 6 Discussion and Conclusion

## 6.1 Discussion

This paper introduces FastLE, a load distribution meta-model method specifically designed for site-specific load
extrapolation. This innovative approach enables the rapid and accurate calculation of 50-year return period extrapolated
loads for various turbine locations within a wind farm, doing so in mere seconds. Utilizing the WTG155-4.55 model with an
Out-of-Plane Bending Moment (OOPBM) load component, FastLE dramatically reduces the simulation time for load
extrapolation from 20 h to just seconds, while maintaining an average Absolute Percentage Error (APE) of only 3.165 %.
This advancement makes it feasible to incorporate extrapolated loads as constraints in wind farm optimization design,
thereby ensuring the structural integrity of wind turbines at specific sites.

The process of load extrapolation for wind turbines consists of numerous steps, each with multiple implementation
approaches. Taking "fitting before aggregation" as an example, peaks are extracted from the time series using three methods:
global maxima, block maxima, and peak over threshold. Various distributions have been applied to these extracted peaks,
including local load distribution functions such as the Weibull, Normal, Rayleigh, and Gumbel distributions. Different
approaches, methods, and parameters can all affect the extrapolated load. Based on relevant literature, this paper selects the
recommended optimal path, which includes using a block size of 30 seconds, block maxima, and a Weibull distribution. The
study primarily investigates the feasibility of a rapid extrapolation method for extreme loads and does not perform a
systematic analysis of importance or parameter sensitivity. During the selection process for the optimal local distribution, the
Kolmogorov-Smirnov goodness-of-fit test was employed for ranking the options. The Weibull distribution emerged with the
highest optimal proportion at 51.986 %, significantly surpassed other distributions. However, the normal distribution also
showed a substantial optimal proportion of 33.833 %, and the Gumbel distribution accounted for 14.181 %. This suggests
that the optimal distribution for specific wind speeds may not always be the Weibull distribution. Nevertheless, due to the
lack of a more suitable method at present, the Weibull distribution was predominantly applied across all wind speeds. This
approach could potentially introduce errors into subsequent load extrapolations. Furthermore, in analyzing the discrepancies
in the extrapolated OOPBM loads for test case 101, it was discovered that the Weibull distribution was not optimal for
certain wind speeds. This finding validates the aforementioned point, as illustrated in the Fig.15. Additionally, inspired by
IEC 61400-1(2019), local distributions at various wind speeds can be represented using a Gaussian mixture model, which
combines multiple Gaussian distributions with different weights. This approach offers a viable avenue for further research in
the field.





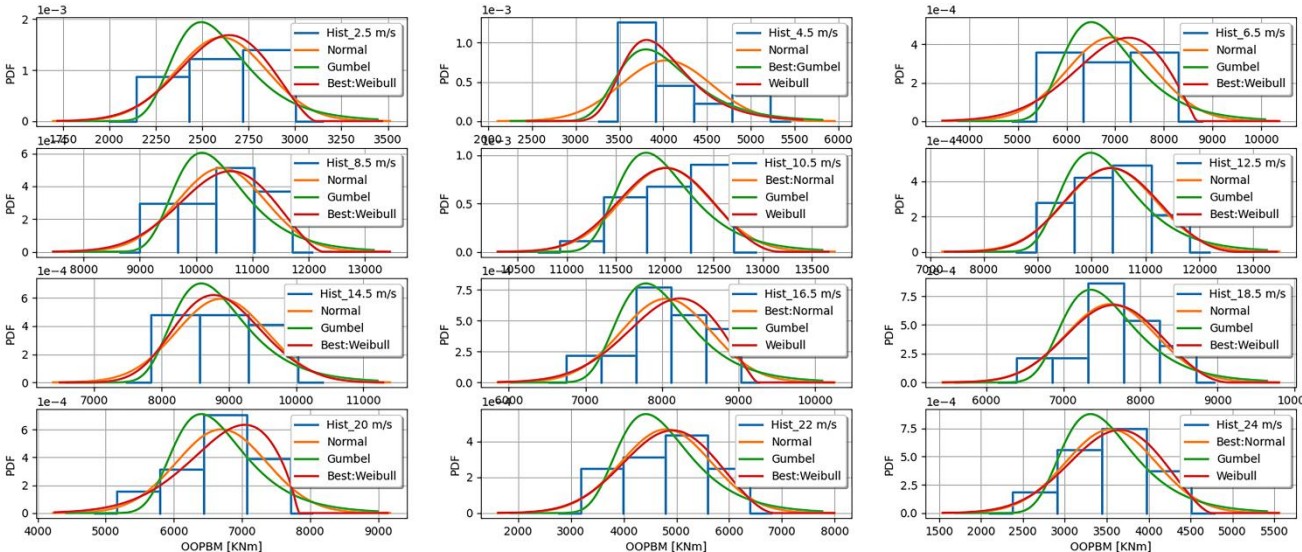

**Figure 15: Optimal local load distributions under different wind speeds for test case 101.**

Another crucial issue is the uncertainty analysis of FastLE. Load extrapolation is fundamentally probabilistic, and its implementation involves several factors that contribute to uncertainty in the results. These factors include the number of load simulation seeds, the methods used for selecting and testing independent samples, the determination of the optimal distribution, and, notably, the training of local distribution parameters within the meta-model. As an example, in the meta-model training process for the local load distribution parameters in Test Case 106, the uncertainty can influence both the local load distribution and the extreme load distribution, as demonstrated in the Fig.16, ultimately affect the load extrapolation results. However, the manner in which these disturbances propagate during the load extrapolation process remains unknown. Related investigations are ongoing, and although this paper does not yet address this aspect, it is undeniable that this issue is of critical importance.





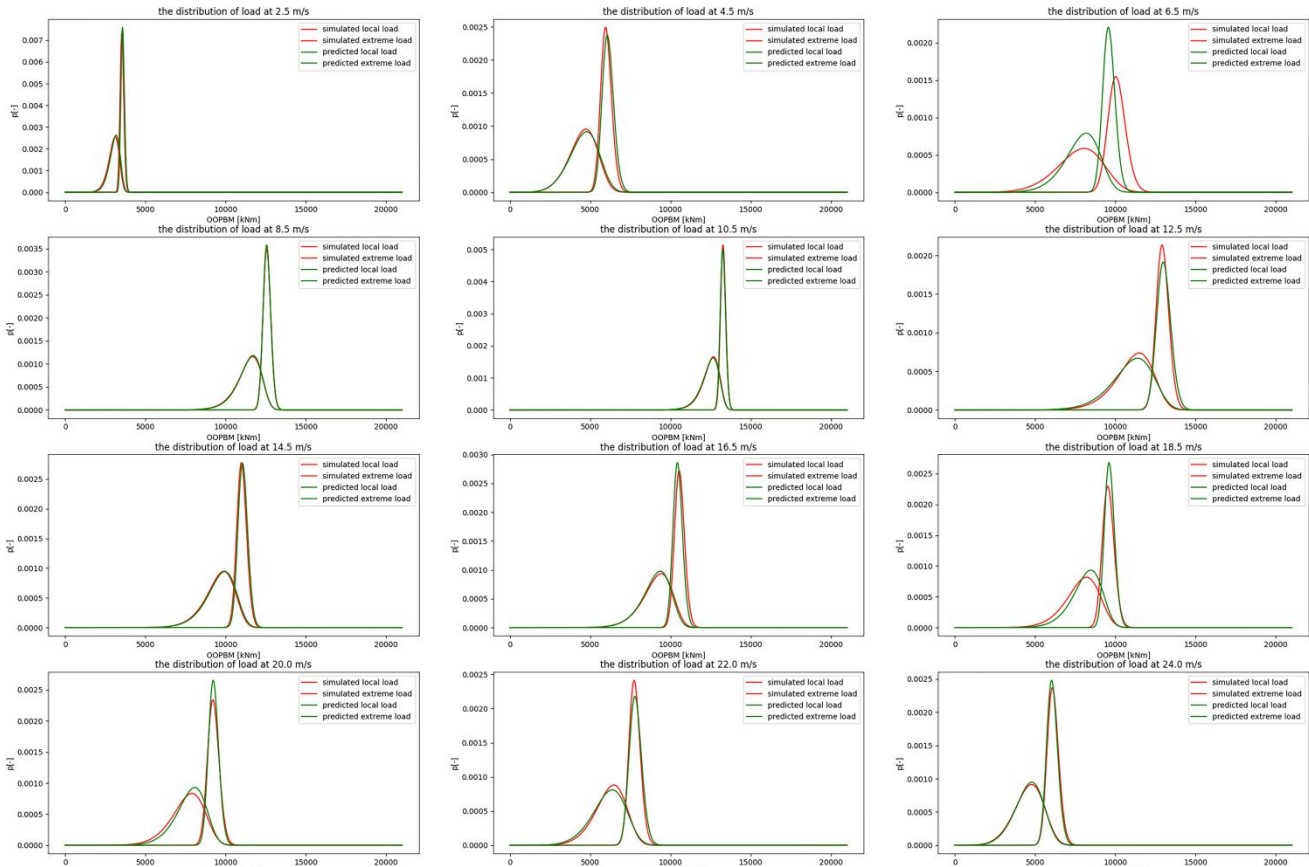

**Figure 16: Local load distributions and extreme load distributions derived from the meta-model and simulated data fitting at various wind speeds for test case 106.**

### 6.2 Conclusion

This paper introduces FastLE, a rapid load extrapolation method based on meta-models and tailored to specific site conditions. By leveraging extensive historical statistical data to determine the range of wind conditions, we employed Monte Carlo sampling to create a training and validation set consisting of 100 cases (including 55800 samples) and 20 test cases (including 11160 samples). The WTG156-4.55 was simulated using Bladed software under 600 s of normal operational conditions, with the Out-of-Plane Bending Moment (OOPBM) chosen as the load for study, thereby generating the data

sources for this research. To ensure the independence of local peak loads, the DcorrX test was introduced, and the Kolmogorov goodness-of-fit test identified the Weibull distribution as optimal for the load. Meta-models for wind to Weibull parameters were trained using Optuna. Finally, FastLE and IEC standard methods were applied to extrapolate the OOPBM for 50-year return periods across 20 test cases. FastLE method achieves an average Absolute Percentage Error (APE) of 3.165 % while reduces computation time to mere seconds,significantly faster than the previous 20 h of IEC



standard method. This advancement enables the use of extrapolated loads as constraints in optimizing the design of each wind turbine in a wind farm, ensuring the structural integrity of wind turbines at specific sites.

Nonetheless, this paper primarily aims to demonstrate the feasibility of the method. There remains a considerable amount of work to be done in the future to further refine and enhance this approach.

(1) Uncertainty analysis is crucial due to the multitude of factors that can introduce uncertainties. These factors encompass
the variability of wind parameters across different ranges and distributions within the database, the selection of independent samples, the determination and fitting of the optimal distribution, the training of meta-models, and the post-processing steps. Ultimately, these uncertainties can impact the reliability of the extrapolated load results.

(2) Sensitivity analysis is vital because various approaches, methods, and parameters employed in the load extrapolation process can influence the outcomes. Thus, conducting a systematic sensitivity analysis is necessary.

(3) Research into the optimal fitting method for local distributions is essential. The statistical findings in this paper indicate that no single distribution can optimally fit every wind speed. Thus, it is crucial to identify a distribution that serves as the best fit for each sample, thereby minimizing the uncertainty of the extrapolated load.

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
