# Peer review of "FastLE: A New Load Extrapolation Method for Site-specific Wind Turbines Using the Load Distribution Meta Model"

_Wind Energy Science, 2025_

## Author Comment (AC3)

**Response to the reviewers**

Dear Reviewer:

We would like to express our sincere gratitude to reviewer for the valuable comments, and the time devoted to review our work. The reviewer gives an accurate summary of our work and brings forward constructive questions. All of the comments are very helpful in improving the quality of our manuscript. We have studied comments carefully and responded to them which are described in detail below.

**Scientific comments:**

**Comment No.1:** The authors mention they use "site-specific" wind parameters, but the wind speed is obtained from the turbine specification, and air density and turbulence intensity are from the literature. In this case, the statistical characteristics of the selected probability distribution fortunately match the measurement. But what if you choose another site? Why not to use the measured statistics directly, e.g. sampling from probabilities in discrete intervals, instead of fitting to a distribution and resampling?

**Response 1:** Thank you for your comment. The proposed method in this paper is applicable to specific turbine models, relying on the wind turbine models used during the generation of the load database. Consequently, the range and values of wind speed must be determined based on the turbine's parameters, specifically from its cut-in wind speed to cut-out wind speed. Furthermore, the ranges for wind parameters such as air density, wind shear, and turbulence intensity are derived from statistical analysis of 541 measured datasets. To maximize coverage with limited samples, the wind parameter distributions were fitted, followed by Monte Carlo (M-C) resampling to densify samples around frequently occurring values, thereby efficiently utilizing high-fidelity load simulation resources. For load assessment at a specific site, the turbine model to be employed is known, thus determining the meta-model trained from the corresponding load database. Given that the value ranges of wind parameters in the database span from P1 to P99, and

the constructed meta-model exhibits certain generalization capabilities, it theoretically enables rapid evaluation of DLC 1.1 for any specific site.

- **Comment No.2:** Figure 5b is not explained. "P-value" is defined neither in the text nor the caption. Does it represent the polulation parameter? Why increasing the block size gives larger P-values if population parameter=0 for independence?

**Response 2:** Thank you for your comment. We will add necessary textual descriptions to all figures in the manuscript.

The p-value in Figure 5(b) is defined in the context of independence testing as the probability of observing the data (or more extreme outcomes) under the assumption that the null hypothesis (i.e., mutual independence) holds true. In essence, the p-value assesses the plausibility of the null hypothesis. For the DcorrX independence test employed in this paper, a smaller p-value provides stronger evidence of an association between the variables, whereas a larger p-value supports the notion of independence.

In load time series, a larger block size corresponds to greater temporal separation between peak loads, resulting in weaker correlations and stronger evidence of peak load independence. Accordingly, in the DcorrX independence test, a p-value approaching 1 indicates greater support for independence.

- **Comment No.3:** While the Weibull distribution emerges as the most common optimal fit (52%), the fact that Normal (34%) and Gumbel (14%) distributions perform better for a substantial proportion of wind speeds suggests that a wind-speed-dependent or mixed fitting approach could improve accuracy. Including a sensitivity study of distribution choice on extrapolated loads would be helpful.

**Response 3:** Thank you for your comment. As the reviewer pointed out, a single fitting distribution indeed cannot accommodate all wind speeds. This issue has been mentioned

in the discussion section of Chapter 6.1 in this paper, as illustrated in Figure 15. We have now begun to explore methods such as adaptive selection of optimal distributions and Gaussian mixture models. Although these approaches have shown improvements in fitting local load distributions, the construction of subsequent meta-models with variable network structures remains a challenging problem that we are still tackling.

● **Comment No.4:** Please clarify how the hyperparameter importance is obtained.

**Response 4:** Thank you for your comment. For the analysis of hyperparameter importance, this paper employs the PED-ANOVA method, as detailed in the literature: https://arxiv.org/abs/2304.10255. This is implemented using the Python class optuna.importance.PedAnovaImportanceEvaluator(), with further details available at: https://optuna.readthedocs.io/en/latest/_modules/optuna/importance/_ped_anova/evaluator.html#PedAnovaImportanceEvaluator.

● **Comment No.5:** Did you consider normalization of MLP input and output parameters? Due to the difference in the order of magnitude between the two outputs, the MLP possibly gives more importance to the paras_1(scale) over paras_0 (shape).

**Response 5:** Thank you for your comment. In this paper, both inputs and outputs have been normalized using the method: $z = (x-\mu)/\sigma$, where $\mu$ is the mean of the training samples, and $\sigma$ is the standard deviation of the training samples. This is implemented via sklearn.preprocessing.StandardScaler(), as detailed at: https://scikit-learn.org/stable/modules/generated/sklearn.preprocessing.StandardScaler.html.

● **Comment No.6:** A figure of the proposed framework (similar to Figure 4) would be helpful to understand the added contribution of this paper compared to the IEC-proposed method.

**Response 6:** Thank you for your positive feedback on the figures. The flowchart of the FastLE method described in this paper is shown below, and it will be presented in the manuscript subsequently.

[Figure]

● **Comment No.7:** Page 14, Line 244, the error is in percentage or true value? Formulate what it meant by "error"?

> **Response 7:** Thank you for your comment. In this paper, the error is defined as error=(predicted value−actual value)/actual value, which will be added to the manuscript subsequently.

● **Comment No.8:** Page 18, Line 276, the time required to generate the training data for FastLE should be taken into account.

> **Response 8:** Thank you for your comment. This time required to generate the training data for FastLE will be included in the relevant section of the paper.

**Technical comments:**

● **Comment No.9:** Reference needed for "It is widely recognized that lower wind speeds contribute minimally to the tails of long-term load distributions."

> **Response 9:** Thank you for your comment. We will add the following references to the paper.
>
> [1]Fogle, J., Agarwal, P., Manuel, L. (2008). Towards an improved understanding of statistical extrapolation for wind turbine extreme loads. Wind Energy, 11(6), 613-635. https://doi.org/10.1002/we.303

● **Comment No.10:** Add the references "To effectively speed up the load extrapolation, this study references certain literature to introduce wind parameters into the Meta model for load components."

> **Response 10:** Thank you for your comment. We will add the following references to the paper.
>
> [1] Dimitrov, N., Kelly, M. C., Vignaroli, A., Berg, J. (2018). From wind to loads: wind turbine site-specific load estimation with surrogate models trained on high-fidelity load

databases. Wind Energy, 21(12), 1384-1402. https://doi.org/10.1002/we.2257

[2] Graf, P. A., Stewart, G., Lackner, M., Dykes, K., Veers, P. (2016). High-throughput computation and the applicability of Monte Carlo integration in fatigue load estimation of floating offshore wind turbines. Wind Energy, 19(5), 921-946. https://doi.org/10.1002/we.1870

● **Comment No.11:** Some abbreviations are used without defining.

**Response 11:** Thank you for your comment. We have checked the abbreviations in the manuscript and added full names for undefined abbreviations.

● **Comment No.12:** The manuscript still needs to be carefully proofread.

**Response 12:** Thank you for your comment. We have reorganized the manuscript and made some minor revisions.

● **Comment No.13:** Figures could be made more self-contained by including parameter definitions and clearer legends.

**Response 13:** Thank you for your comment. We will add necessary textual descriptions to all figures in the manuscript.

Once again, thank you very much for the constructive comments and suggestions which would help us in depth to improve the quality of the manuscript. We will try our best to improve the manuscript. Please feel free to contacts with any questions.

Kind regards,

Pengfei Zhang

---

## Author Response (AR1)

Response Letter for 'FastLE: A Rapid
Load Extrapolation Method for Wind
Turbines at Specific Sites Using the
Load Distribution Meta Model'

Journal Title: Wind Energy Science

Manuscript Number: WES-2025-39

Title of Paper: FastLE: A Rapid Load Extrapolation Method for

Wind Turbines at Specific Sites Using the Load Distribution Meta

Model

**Dear Editor and Reviewers:**

We sincerely appreciate the opportunity to publish our work in **Wind Energy Science** and are deeply grateful to the editor and reviewers for their time, expertise, and constructive feedback, which have greatly enhanced the quality of our manuscript. We are particularly encouraged by their recognition of our work's significance and have revised the manuscript based on the editor-in-chief's comments. The "**Author contribution**" section has been added as requested. All main changes are highlighted in colored to facilitate an ease review of the paper.

**1. Response to the comments of Reviewer #EC1**

Dear authors, thank you for submitting this interesting paper. I have a few comments which I hope will be complementary to the upcoming reviews.

The authors mention an impressive dataset of 541 meteorological towers in a specific region. Some more details would be relevant in order to understand if the data are comparable – i.e., how does the terrain differ among the various met mast locations, are the measurement heights the same, are the instruments the same (cup anemometers, sonic anemometers, lidars)?

The IEC 61400, ed. 4 standard allows several different approaches to extrapolation, including avoiding the extrapolation altogether by introducing a higher safety factor. It will be useful if the authors could study/compare these different extrapolation approaches in the context of their proposed methodology.

One significant challenge in the "fitting before aggregation" method is that the distribution fitting on a few values is not very robust, and a few outliers or bad fits can distort the aggregated result. It would be good to check the confidence in the aggregated distribution predictions – for example by doing multiple local distribution fits by bootstrapping the block maxima.

There is a dependency between the shape and scale parameters in a Weibull distribution fit (if you choose a value of one parameter, it will define what is the value of the other parameter that best represents the data set). Therefore, fitting separate

meta models for the scale and shape parameters of the Weibull distribution may limit the accuracy of the results. In the current manuscript, it doesn't get clear if the authors fit one single MLP model with two outputs, or two separate models? Please discuss.

First of all, we would like to express our sincere gratitude to Reviewer #EC1 for the valuable comments, and the time devoted to review our work. The reviewer brings forward constructive questions. All of the comments are very helpful in improving the quality of our manuscript. We have carefully referred to each of the comments and made changes accordingly. The main corrections in the paper and the responds to the reviewer's comments are as flowing:

**1.1 Reviewer #EC1, Comment No.1:**

The authors mention an impressive dataset of 541 meteorological towers in a specific region. Some more details would be relevant in order to understand if the data are comparable – i.e., how does the terrain differ among the various met mast locations, are the measurement heights the same, are the instruments the same (cup anemometers, sonic anemometers, lidars)?

**Response 1.1:** Thank you for your comment. The areas of meteorological towers used in this paper are mainly in North China. Most of the meteorological towers are located in the terrain of plains and hills, which are judged as L, M, and H classes in accordance with the terrain in the IEC61400-1:2019 Chapter 11.2 with the approximate proportions of 50%, 30%, and 20%.

The lowest height of these meteorological towers installed with wind speed and direction sensors is 10 m, the highest height is between 70 m and 140 m, and using the wind shear exponent to uniformly extrapolate to a height of 100 m.

The anemometers used in the meteorological towers are cup anemometers, no ultrasonic or LiDAR.

The principle of using the above database is to cover as wide a range of wind parameters as possible, making the model more widely applicable. We add the above description in the manuscript. The revised parts are in the section 2, as follows:

The data utilized for extrapolation methods is derived from time series simulations of the turbine

operating across a specified wind range. The areas of meteorological towers used in this paper are mainly in North China. Most of the meteorological towers are located in the terrain of plains and hills, which are judged as L, M, and H classes in accordance with the terrain in the IEC61400-1:2019 Chapter 11.2 with the approximate proportions of 50%, 30%, and 20%. The lowest height of these meteorological towers installed with wind speed and direction sensors is 10 m, the highest height is between 70 m and 140 m, and using the wind shear exponent to uniformly extrapolate to a height of 100 m. The anemometers used in the meteorological towers are cup anemometers.

**1.2 Reviewer #EC1, Comment No.2:**

The IEC 61400, ed. 4 standard allows several different approaches to extrapolation, including avoiding the extrapolation altogether by introducing a higher safety factor. It will be useful if the authors could study/compare these different extrapolation approaches in the context of their proposed methodology.

**Response 1.2:** Thank you for your comment. This is a very good suggestion, and we believe that the reason why the IEC standard allows for the existence of different extrapolation methods with a high safety factor is that each method has a rationale and the truth value cannot be verified to a certain extent, and in this case a high safety factor can only be used to ensure safety.

The core objective of this paper is to explore the feasibility of the technical approach. Comparisons between different technical methods and uncertainty analysis will not be addressed in this paper.

We have identified the technical route of "fitting before aggregation" through previous research, and in the test case we have only compared the results under this route and called it the IEC method. The aggregation before fitting and the inverse first-order reliability method (IFORM) are not compared in the current manuscript. In fact, this work we are in progress because we also realize the differences in the results of the different methods and the importance of uncertainty analysis in the extrapolation of ultimate loads.

**1.3 Reviewer #EC1, Comment No.3:**

One significant challenge in the "fitting before aggregation" method is that the distribution fitting on a few values is not very robust, and a few outliers or bad fits can

distort the aggregated result. It would be good to check the confidence in the aggregated distribution predictions – for example by doing multiple local distribution fits by bootstrapping the block maxima.

Response 1.3: Thank you for your comment. The problem you mentioned is indeed the challenge we face in this method. In the process of local distribution fitting, we introduced the Normal, Log-normal, Gumbel and Weibull distributions as candidate distributions, and through the fitting test, we found that none of these distributions can fully satisfy all the samples. As described in Section 4.2, in order to ensure the consistency of the MLP model output, we chose the Weibull distribution, which fits the most samples better. This also leads to a deviation of the local distribution from reality for some wind speeds, refer to the 6.1 Discussion, which perturbs the load extrapolated results. In fact, we have seen in the literature that the use of Gaussian mixture model gives better results than distributions such as Weibull to avoid and reduce outliers or bad fits, now we are trying in this way. And we will also try by doing multiple local distribution fits and by bootstrapping the block maxima.

**1.4 Reviewer #EC1, Comment No.4:**

There is a dependency between the shape and scale parameters in a Weibull distribution fit (if you choose a value of one parameter, it will define what is the value of the other parameter that best represents the data set). Therefore, fitting separate meta models for the scale and shape parameters of the Weibull distribution may limit the accuracy of the results. In the current manuscript, it doesn't get clear if the authors fit one single MLP model with two outputs, or two separate models? Please discuss.

**Response 1.4:** Thank you for your comment. Actually we have used one single MLP model with three outputs. But due to the Weibull distribution parameter loc=0 we have used, it can also be considered as two outputs. We described it in detail in the manuscript.

The input of the Local Peaks Distribution Meta-Model includes wind speed, its corresponding turbulence intensity, wind shear, air density, inflow angle, and yaw misalignment (configured according to the IEC standard with values of -8, 0, and 8 degrees). The output comprises the shape parameter and scale parameter of the Weibull distribution (with the location parameter set to 0).

Additionally, normalization is applied based on the training samples, using the method shown in the following equation:

$$z = \frac{x - \mu}{\sigma} \tag{12}$$

where  $\mu$  is the mean of the training samples, and  $\sigma$  is the standard deviation of the training samples.

**2. Response to the comments of Reviewer #RC1**

The paper deals with a rapid evaluation of the extreme loads using extrapolation methods, currently the extrapolation requires a significant number of simulations to provide sufficient samples of extreme loads in order to perform the extrapolation procedure. The manuscript uses a machine learning based meta model to accelerate this process while providing extrapolation result with uncertainties comparable to those using aeroelastic simulations. One important question that needs to be clarified is what is the value of the meta model compared to the many load surrogate models that are available in the literature. After all the main time saving is coming from the meta load model which is essentially another load surrogate model.

1- when comparing the time saving, how would the authors account the time and efforts needed to produce the data needed to train the meta model. since this would be necessary each time the turbine model or turbine properties have been changes, which is often the case in the design iteration phase. Normally the 50 years return value for extreme load is a design value based on generic wind class or site specific value for certain class of sites, for example typhoon or hurricane affected area. It is usually not needed to perform load extrapolation for each of the wind turbine in a wind farm. Once it has been identified which turbine in the wind farm has the highest extreme loads, one needs only to perform the load extrapolation for the worst case. It is rather unlikely that optimization for extreme loads will be performed for every single turbine. Moreover, it is not clear from the beginning of the design, whether fatigue or extreme load will be the design driver. Therefore, the usefulness and time saving should be considered with these points in mind.

2- In page three, line 84, the word inflow angle is mentioned. In this case, it is

referred to the yaw angle between the rotor plane with the incoming wind, that is, the yaw misalignment angle, if the reviewer understands it correctly. Inflow angle is used in the aerodynamics mainly for the angle of the velocity triangle at the airfoil, between the tangential velocity caused by the rotation of the rotor and the incoming wind velocity. The use of the word inflow angle can cause some confusion as this is not used normally in this context.

- 3- Page 4. which is the shear model used and how is the shear value defined, please elaborate.
- 4- Figure 1, the distribution of the air density looks bi-modal, when sampling the distribution, did the authors take the empirical distribution or the fitted bi-modal distribution
- 5- Table 1 why is the inflow angle changes from -0.78 to 13.464 degrees (there is no need to go beyond the first digit for this angle, the turbine yaw controller is not that precise), what about the variation in the negative angle, the loading on the wind turbine is not symmetrical around the yaw angle, negative and positive yaw angles can produce very different loads.

**6- Table 2 change RMP to RPM**

- 7- page 7 what is the definition of In plane and out of plane bending moment here. It looks like the authors is using the flapwise bending moment and not the out of plane bending moment of the blade. Once the blade starts pitching after reaching the rated wind speed, the the OOP bending moment and flapwise bending moment are no longer the same.
- 8- Equation 6, this equation assumes that the 10 minutes wind speeds are independent, which is clearly not the case.
- 9- page 9, line 171, the authors divided the data into three categories, high wind speed range above 10 m/s, low wind speed range below 10 m/s and full wind speed range, which wind speed would be full wind speed range have?
- 10- Figure 6 why are the log-normal performed so poorly in QQ plot
- 11- Table 3, there is not need to have numbers with 9 digits after the decimal point, there are a lot of uncertainties

- 12- Figure 10, how ar ehte importance of the hyperparameters determined?
- 13- page 9 line 176, so if the low wind speeds contribute so little to the tail of the distribution, then why simulate them at all.
- 14- instead of local distribution, maybe it is better to refer them as local maxima, or local peaks distribution.
- 15- Table 5, the simulation time is 600seconds, what about the transient at the beginning of the simulation, are they removed?

We would like to express our sincere gratitude to Reviewer #RC1 for the valuable comments, and the time devoted to review our work. The reviewer brings forward constructive questions, as well as the important guiding significance to our researches. We have studied comments carefully and responded to them which are described in detail below.

**2.1 Reviewer #RC1, Comment No.1:**

For the question that needs to be clarified is what is the value of the meta model compared to the many load surrogate models that are available in the literature, our specific explanation is below.

**Response 2.1:** Thank you for your comment.**

- 1. In this paper, a Meta-model is established for the parameters of the local load distribution at each wind speed, whereas existing load surrogate models primarily focus on single load values.
- 2. The Meta mode mentioned in this paper is constructed based on MLP (Multi-layer Perceptron), which essentially focuses on its ability to capture complex nonlinear relationships, provide reliable results, and reduce the number of training samples. Compared to the traditional load surrogate models (Linear Regression, Polynomial Regression, Gaussian Process and Response Surface Regression), it performs better when dealing with complex systems, especially when data and computational resources are limited. In fact, we have also used the above method to try to compare and selected MLP based on the accuracy of the results, but we have not expressed it in the paper due to space limitation.

However, the optimal approach must be determined based on the specific application scenario, data characteristics, and available computational resources[1]. Future research should further investigate the potential of multi-fidelity data fusion and deep learning techniques to improve model accuracy and robustness.

**References:**

[1] Angione C, Silverman E, Yaneske E, (2022) Using machine learning as a surrogate model for agent-based simulations. PLoS ONE, 17(2): e0263150.

**2.2 Reviewer #RC1, Comment No.2:**

1-when comparing the time saving, how would the authors account the time and efforts needed to produce the data needed to train the meta model. since this would be necessary each time the turbine model or turbine properties have been changes, which is often the case in the design iteration phase. Normally the 50 years return value for extreme load is a design value based on generic wind class or site specific value for certain class of sites, for example typhoon or hurricane affected area. It is usually not needed to perform load extrapolation for each of the wind turbine in a wind farm. Once it has been identified which turbine in the wind farm has the highest extreme loads, one needs only to perform the load extrapolation for the worst case. It is rather unlikely that optimization for extreme loads will be performed for every single turbine. Moreover, it is not clear from the beginning of the design, whether fatigue or extreme load will be the design driver. Therefore, the usefulness and time saving should be considered with these points in mind.

**Response 2.2:** Thank you for your comment. The load extrapolation methods mentioned in this paper are mainly used for site suitability assessment and will not be used for iteration in WTG development and design. Here is a more specific explanation.

- 1. The model used for generating the load simulation database is either a finalized model or a wind turbine model that has obtained DA/TC certification.
- 2. According to IEC 61400-1:2019 Annex B, for site suitability assessment, the following ultimate design load cases shall be assessed as minimum: DLC 1.1, DLC 1.3, DLC 6.1, and DLC 6.2. If the design load cases for the standard classes are

adequate, no further evaluations need to be performed. The DLC1.1 is also required for site-specific calculations. The "adequate" scenario is only qualitatively described in the standard, it is often difficult to prove it to an independent third-party certification body during the SSDA certification process, so it is necessary to perform the DLC1.1 for site-specific projects.

3. The worst case is a common practice in previous years, but nowadays it has become a mainstream trend to strike a balance between economy and conservatism, so wind turbines in wind farms with complex terrain are usually divided into groups to replace the worst case. With the rapid development of the wind power industry, a top OEM will do at least a thousand wind farm site suitability assessments per year, and the amount of computation and time required for the DLC1.1 is very large. In addition, it is the probability density of each wind speed bin and the corresponding turbulence intensity that plays a role in the DLC1.1 case, the worst case is extremely conservative in some cases, which is a common situation in China. And when doing SSDA certification, the independent third-party certification bodies usually require proof of the vague description of the worst case. Using the methodology mentioned in this paper, each turbine can be quickly evaluated to determine the worst case, and then using Bladed/FAST/HAWC2 can be performed, which is a common practice in the industry.

In summary, the proposed FASTLE method demonstrates significant potential for site-specific preliminary load assessment, grouping, and worst-case selection. Moreover, it offers substantial reductions in computational cost and processing time compared to conventional approaches.

**2.3 Reviewer #RC1, Comment No.3:**

2-In page three, line 84, the word inflow angle is mentioned. In this case, it is referred to the yaw angle between the rotor plane with the incoming wind, that is, the yaw misalignment angle, if the reviewer understands it correctly. Inflow angle is used in the aerodynamics mainly for the angle of the velocity triangle at the airfoil, between the tangential velocity caused by the rotation of the rotor and the incoming wind

velocity. The use of the word inflow angle can cause some confusion as this is not used normally in this context.

Response 2.3: Thank you for your comment. The inflow angle referenced in this paper is derived from IEC 61400-15-1:2025 (Section 5.4) and illustrated in Fig. 1. It is critical to note that this parameter does not represent the yaw misalignment angle but aligns with the definition of flow inclination as specified in IEC 61400-1:2019. While DNV-ST-0473 also employs inflow angle to describe flow inclination, this usage may lead to ambiguity with the inflow angle defined in Blade Element Momentum (BEM) theory. To avoid confusion, we adopted the flow inclination consistently throughout the paper. The corresponding modifications have been marked in the manuscript, as shown below:

Aside from wind speed, the primary wind parameters affecting the loads on wind turbines include air density, turbulence intensity, wind shear, and flow inclination...

The load time series simulation is run for different wind speeds, from cut-in to cut-out wind speeds, under normal power production conditions. This simulation is based on the site-specific wind parameters which include air density, turbulence intensity at different wind speeds, wind shear, flow inclinations, etc.

...For any given wind speed  $V_i$ , corresponding turbulence intensity  $TI_i$ , wind shear  $\alpha_i$ , and flow inclination  $I_i$  are used as inputs for the Local Load Distribution Meta Model.

**5.4 Inflow angle**

An inflow angle for each wind direction sector (i) based on measured and/or simulated values from a validated flow model shall be calculated by using the following equation:

$$\varphi_i = \tan^{-1} \left( \frac{v_z}{v_{xy}} \right) \tag{1}$$

If no site measurements or simulations are available, the <a href="inflow">inflow</a> angle may be estimated based on terrain slope according to IEC 61400-1:2019, 11.9.2.

To calculate the omni-directional inflow angle either a frequency or energy-weighted mean shall be performed.

Fig. 1 IEC 61400-15-1:2025 chapter 5.4

**3.5.3 Basic wind parameters for design**

**3.5.3.1** The basic site-specific wind parameters which shall be determined as input to the design are listed below:

- $^{-}$  long-term mean wind speed at hub height  $V_{ave}$  and wind speed distribution, see [3.5.5]
- wind direction distribution (wind rose) per wind speed bin and accumulated, see [3.5.6]
- wind shear and veer, see [3.5.7]
- mean ambient turbulence intensity and standard deviation of the turbulence intensity at hub height as a function of wind speed and wind direction, see [3.5.10]
- $^-$  reference wind speed  $V_{ref}$ , which is defined as the 50-year 10-min mean value  $V_{50}$ , and extreme 50-year gust wind speed  $V_{e50}$  (50-year 3-sec gust)], see [3.5.11].

3.5.3.2 For onshore projects only:

inflow angle. If a significant part of the energy comes from a sector with negative inflow angle or with more than 8° inflow angle, the directional dependency of the inflow angle shall be considered.

Standard — DNV-ST-0437. Edition May 2024 Loads and site conditions for wind turbines

Fig. 2 DNV-ST-0473

**2.4 Reviewer #RC1, Comment No.4:**

3-Page 4. which is the shear model used and how is the shear value defined, please elaborate.

**Response 2.4:** Thank you for your comment. The power-law shear profile is used model in this paper. We added a description to the paper.

$$V(z) = V(z_r) \left(\frac{z}{z_r}\right)^{\alpha} \tag{1}$$

V(z) is the wind speed at height z, z is the height above ground,  $z_r$  is a reference height above ground used for fitting the profile,  $\alpha$  is the wind shear (or power law) exponent.

**2.5 Reviewer #RC1, Comment No.5:**

4-Figure 1, the distribution of the air density looks bi-modal, when sampling the distribution, did the authors take the empirical distribution or the fitted bi-modal distribution?

**Response 2.5:** Thank you for your comment. The air density distribution is obtained using the Kernel Smoothing method based on a large amount of test data, the reference is as follow.

Reference:

[2] M. P. Wand & M. C. Jones Kernel Smoothing Monographs on Statistics and Applied Probability Chapman & Hall, 1995.

**2.6 Reviewer #RC1, Comment No.6:**

5-Table 1 why is the inflow angle changes from -0.78 to 13.464 degrees (there is no need to go beyond the first digit for this angle, the turbine yaw controller is not that precise), what about the variation in the negative angle, the loading on the wind turbine is not symmetrical around the yaw angle, negative and positive yaw angles can produce very different loads.

**Response 2.6:** Thank you for your comment. In fact, for yaw misalignment angle setting, this paper uses equal numbers -8, 0, 8(refer to The table 5). The inflow angle here refers to the flow inclination.

**2.7 Reviewer #RC1, Comment No.7:**

6-Table 2 change RMP to RPM.

**Response 2.7:** Thank you for your comment. We have made changes in the manuscript.

|    | parameters√         | unit₽ | value⊬ | * |
|----|---------------------|-------|--------|---|
|    | cut-in-wind-speed   | m/s₊  | 2.5₽   | 4 |
|    | cut-out-wind-speed  | m/s₊  | 24₽    | * |
|    | rated wind speed    | m/s₊  | 10.8₽  | * |
|    | Rotor rated speed   | rpm₽  | 9.5₽   | 4 |
| Į. | Rotor speed range ↓ | rpm₄  | 6~9.5+ | * |

**2.8 Reviewer #RC1, Comment No.8:**

7-Page 7 what is the definition of In plane and out of plane bending moment here. It looks like the authors is using the flapwise bending moment and not the out of plane bending moment of the blade. Once the blade starts pitching after reaching the rated wind speed, the the OOP bending moment and flapwise bending moment are no longer the same.

**Response 2.8:** Thank you for your comment. In this paper we used the blade coordinate system from the GL2012, as shown in Fig. 3. The MYB is out of plane

bending moment and MXB is in plane bending moment, which was described in the paper.

In accordance with wind turbine design standards, the analysis of load extrapolation concerning the structural integrity must include at least the computation of extreme values for the blade root in-plane bending moment(IPBM), out-of-plane bending moment(OOPBM), and tip deflection, as shown in Fig.3. The IPBM and OOPBM is described using the blade coordinate system from the IV-Part 1 GL(2012), as shown in Fig. 4. The MYB is out of plane bending moment and MXB is in plane bending moment. This paper focuses on the methodological exposition, so the out-of-plane bending moment at the blade root will be analyzed as an example.

Fig. 4 Coordinate system of wind turbine blade

Germanischer Lloyd Industrial Services GmbH: IV-Part 1 GL: Guideline for the Certification of Wind Turbines, 2012.

**2.9 Reviewer #RC1, Comment No.9:**

8-Equation 6, this equation assumes that the 10 minutes wind speeds are independent, which is clearly not the case.

**Response 2.9:** Thank you for your comment. The equation 6 is the calculation of the exceedance probability of the 50-year extreme load. The probability of the 50-year load is approximately  $3.8 \times 10^{-7}$ , is consistent with the definition in IEC 61400-1:2019.

**2.10 Reviewer #RC1, Comment No.10:**

9-Page 9, line 171, the authors divided the data into three categories, high wind speed range above 10 m/s, low wind speed range below 10 m/s and full wind speed range, which wind speed would be full wind speed range have?

**Response 2.10:** Thank you for your comment. The full wind speed means Cut-in wind speed to cut-out wind speed, which includes 2.5m/s~24m/s.

**2.11 Reviewer #RC1, Comment No.11:**

10-Figure 6 why are the log-normal performed so poorly in QQ plot?

Response 2.11: Thank you for your comment. The distribution of the extracted peak loads at a certain wind speed does not obey the log-normal distribution, which is confirmed by using the Chi-Squared test and the Kolmogorov -Smirnov goodness-offit test to test the log-normal. Therefore, when QQ-plot was used to visualize the presentation, the log-normal performance was not good as shown in Figure 6. In fact, at that time, when we used openturns (python package) to do distribution fitting work and got very poor log-normal distribution fitting results, we used the same data and switched to scipy (python package) to do log-normal fitting and Chi-Squared test, and found that the results did not change.

**2.12 Reviewer #RC1, Comment No.12:**

11-Table 3, there is not need to have numbers with 9 digits after the decimal point, there are a lot of uncertainties.

**Response 2.12:** Thank you for your comment. We have made changes in the manuscript. All relevant numbers are retained to 4 digits.

| The pass rate of | of 55800 sample | S∉                | All-samples | low wind speed (<= 10 m·s -1 ) | high wind speed (>10 m·s -1 ) v |
|------------------|-----------------|-------------------|-------------|-------------------------------------------|-------------------------------------------------------|
|                  | Weibull         | TRUE₽             | 99.958 %    | 99.900 %                                  | 100.000 %                                             |
|                  |                 | p-value(1.%level) | 0.6739↓     | 0.6570                                    | 0.6859                                                |
|                  | NT-MARKET       | TRUE₽             | 99.903 %    | 99.767 •% ↔                               | 100.000 %                                             |
| Chi-Squared      | Normal.         | p-value(1-%level) | 0.6528+     | 0.63334                                   | 0.6668                                                |
| test.            | Gumbel.         | TRUE↓             | 99.944*%    | 99.867 %                                  | 100.000 %                                             |
|                  | Gumber          | p-value(1.%level) | 0.5824      | 0.52324                                   | 0.6247                                                |
|                  | Log-Normal.     | TRUE⊬             | 0.028 %     | 0.067 %                                   | 0.000 %                                               |
|                  |                 | p-value(1-%level) | 1.2156×105  | 2.8656×105+                               | 3.7033×10 7 €                              |

**2.13 Reviewer #RC1, Comment No.13:**

12-Figure 10, how are the importance of the hyperparameters determined?

**Response 2.13:** Thank you for your comment. Based on the Optuna (python package) using Fanova Importance Evaluator to implement it, see reference for a description of the methodology.

**Reference:**

[3] Frank H, Holger H, Kevin L. An Efficient Approach for Assessing Hyperparameter

Importance, Proceedings of the 31st International Conference on Machine Learning, PMLR 32(1): 754-762, 2014.

**2.14 Reviewer #RC1, Comment No.14:**

13-Page 9 line 176, so if the low wind speeds contribute so little to the tail of the distribution, then why simulate them at all.

**Response 2.14:** Thank you for your comment. According to the practical experience, the low wind speed contributes extremely little to the 50-year extreme load extrapolation, but it does not mean that there is none at all. and the IEC 61400-1:2019 (Section 7.6.2.2) has made a requirement for the simulation of low wind speeds, as shown in fig. 4, so this paper also carries out the simulation analysis of low wind speeds.

**7.6.2.2 Partial safety factors for loads**

For DLC 1.1, a characteristic value of load shall be determined by a statistical analysis of the extreme loading that occurs for normal design situations and shall correspond to one of the following alternatives.

- a) The characteristic value is obtained as the largest (or smallest) among the average values of the 10 min extremes determined for each wind speed in the given range, multiplied by 1,35. This method can only be applied for the calculation of the blade root in-plane moment and out-of-plane moment and tip deflection.
- b) The characteristic value is obtained as the largest (or smallest) among the 99th percentile (or 1st percentile in the case of minima) values of the 10 min extremes determined for each wind speed in the given range, multiplied by 1,2.
- c) The characteristic value is obtained as the value corresponding to a 50 year return period, based on load extrapolation methods, considering the wind speed distribution given in 6.3.2.1 and the normal turbulence model in 6.3.2.3. Guidance about load extrapolation is given in Annex G.

The design load will be then obtained by multiplying the characteristic loads according to any of these alternatives by the partial safety factor for DLC 1.1 defined in Table 3.

For all three alternatives above, data used in the statistical analysis shall be extracted from time series of turbine simulations of at least 10 minutes in length over the operating range of the turbine for DLC 1.1. A minimum of 15 simulations is required for each wind speed from  $(V_r-2 \text{ m/s})$  to cut-out, and six simulations are required for each wind speed below  $(V_r-2 \text{ m/s})$ . When extracting data, the designer shall consider the effect of independence between peaks on the statistical analysis and minimize dependence when possible. For guidance on dependency checks, see Annex G.

For load cases with specified deterministic wind field events, the characteristic value of the load shall be the worst case computed transient value. If more simulations are performed at a given wind speed, representing the rotor azimuth, the characteristic value for the load case is taken as the average value of the worst case computed transient values at each azimuth. Guidance for the derivation of the contemporaneous load can be found in Annex I. When turbulent inflow is used, the mean value among the worst case computed loads for different

**2.15 Reviewer #RC1, Comment No.15:**

14-instead of local distribution, maybe it is better to refer them as local maxima, or local peaks distribution.

**Response 2.15:** Thank you for your comment. We will be changed to local peaks distribution.

...and the local peaks distribution is fitted to the peaks at each wind speed.

The local peaks distribution function of the extracted peak loads is fitted using the maximum likelihood method.

The local peaks distribution function of extracted peak loads is typically modeled using a Weibull distribution[5,27].

During the selection process for the optimal local peaks distribution, the Kolmogorov-Smirnov goodness-of-fit test was employed for ranking the options.

...local peaks distributions at various wind speeds can be represented using a Gaussian mixture model,

...the training of local peaks distribution parameters within the meta-model.

Research into the optimal fitting method for local peaks distributions is essential.

**2.16 Reviewer #RC1, Comment No.16:**

15-Table 5, the simulation time is 600seconds, what about the transient at the beginning of the simulation, are they removed?

**Response 2.16:** Thank you for your comment. In fact, when bladed was used for the simulation, the simulation duration was set to 650 seconds, and the latter 600 seconds was used for the data output, which will be supplemented in the paper.

Simulation time and The simulation duration was set to 650 s, and the latter 600 s was used for the data output.

**3. Response to the comments of Reviewer #RC2**

The manuscript presents FastLE, a meta-model—based method for rapid extrapolation of extreme wind turbine loads, aiming to reduce the computational burden of IEC 61400-1 site-specific load analysis. The work addresses a relevant and underexplored problem, and the results indicate high agreement with the IEC reference method for simulated cases.

However, the following issues should be addressed before this work can be considered for publication:

**Scientific comments**

The authors mention they use "site-specific" wind parameters, but the wind speed is obtained from the turbine specification, and air density and turbulence intensity are from the literature. In this case, the statistical characteristics of the selected probability distribution fortunately match the measurement. But what if you choose another site? Why not to use the measured statistics directly, e.g. sampling from probabilities in discrete intervals, instead of fitting to a distribution and resampling? Figure 5b is not explained. "P-value" is defined neither in the text nor the caption. Does it represent the polulation parameter? Why increasing the block size gives larger P-values if population parameter=0 for independence?

While the Weibull distribution emerges as the most common optimal fit (52%), the fact that Normal (34%) and Gumbel (14%) distributions perform better for a substantial proportion of wind speeds suggests that a wind-speed-dependent or mixed fitting approach could improve accuracy. Including a sensitivity study of distribution choice on extrapolated loads would be helpful.

Please clarify how the hyperparameter importance is obtained.

Did you consider normalization of MLP input and output parameters? Due to the difference in the order of magnitude between the two outputs, the MLP possibly gives more importance to the paras 1(scale) over paras 0 (shape).

A figure of the proposed framework (similar to Figure 4) would be helpful to understand the added contribution of this paper compared to the IEC-proposed method.

Page 14, Line 244, the error is in percentage or true value? Formulate what it meant by "error"?

Page 18, Line 276, the time required to generate the training data for FastLE should be taken into account.

**Technical comments**

Reference needed for "It is widely recognized that lower wind speeds contribute minimally to the tails of long-term load distributions."

Add the references "To effectively speed up the load extrapolation, this study references certain literature to introduce wind parameters into the Meta model for load

components."

Some abbreviations are used without defining.

The manuscript still needs to be carefully proofread.

Figures could be made more self-contained by including parameter definitions and clearer legends.

We would like to express our sincere gratitude to Reviewer #RC2 for the valuable comments, and the time devoted to review our work. The reviewer brings forward constructive questions, as well as the important guiding significance to our researches. We have studied comments carefully and responded to them which are described in detail below.

**Scientific comments**

**3.1 Reviewer #RC2, Comment No.1:**

The authors mention they use "site-specific" wind parameters, but the wind speed is obtained from the turbine specification, and air density and turbulence intensity are from the literature. In this case, the statistical characteristics of the selected probability distribution fortunately match the measurement. But what if you choose another site? Why not to use the measured statistics directly, e.g. sampling from probabilities in discrete intervals, instead of fitting to a distribution and resampling?

Response 3.1: Thank you for your comment. The proposed method in this paper is applicable to specific turbine models, relying on the wind turbine models used during the generation of the load database. Consequently, the range and values of wind speed must be determined based on the turbine's parameters, specifically from its cut-in wind speed to cut-out wind speed. Furthermore, the ranges for wind parameters such as air density, wind shear, and turbulence intensity are derived from statistical analysis of 541 measured datasets. To maximize coverage with limited samples, the wind parameter distributions were fitted, followed by Monte Carlo (M-C) resampling to densify samples around frequently occurring values, thereby efficiently utilizing high-fidelity load simulation resources. For load assessment at a specific site, the turbine model to be employed is known, thus determining the meta-model trained

from the corresponding load database. Given that the value ranges of wind parameters in the database span from P1 to P99, and the constructed meta-model exhibits certain generalization capabilities, it theoretically enables rapid evaluation of DLC 1.1 for any specific site.

**3.2 Reviewer #RC2, Comment No.2:**

Figure 5b is not explained. "P-value" is defined neither in the text nor the caption. Does it represent the polulation parameter? Why increasing the block size gives larger P-values if population parameter=0 for independence?

**Response 3.2:** Thank you for your comment. The p-value in Figure 6(b) is defined in the context of independence testing as the probability of observing the data (or more extreme outcomes) under the assumption that the null hypothesis (i.e., mutual independence) holds true. In essence, the p-value assesses the plausibility of the null hypothesis. For the DcorrX independence test employed in this paper, a smaller p-value provides stronger evidence of an association between the variables, whereas a larger p-value supports the notion of independence.

In load time series, a larger block size corresponds to greater temporal separation between peak loads, resulting in weaker correlations and stronger evidence of peak load independence. Accordingly, in the DcorrX independence test, a p-value approaching 1 indicates greater support for independence. The relevant description in the paper is as follow:

Similar to other independence tests, p-values are employed to represent the statistical probability under the premise that two variables are independent (the population parameter equals zero), serving as evidence for the mutual independence between the variables.

**3.3 Reviewer #RC2, Comment No.3:**

While the Weibull distribution emerges as the most common optimal fit (52%), the fact that Normal (34%) and Gumbel (14%) distributions perform better for a substantial proportion of wind speeds suggests that a wind-speed-dependent or mixed fitting approach could improve accuracy. Including a sensitivity study of distribution choice on extrapolated loads would be helpful.

Response 3.3: Thank you for your comment. As the reviewer pointed out, a single

fitting distribution indeed cannot accommodate all wind speeds. This issue has been mentioned in the discussion section of Chapter 6.1 in this paper, as illustrated in Figure 17. We have now begun to explore methods such as adaptive selection of optimal distributions and Gaussian mixture models. Although these approaches have shown improvements in fitting local load distributions, the construction of subsequent meta-models with variable network structures remains a challenging problem that we are still tackling.

**3.4 Reviewer #RC2. Comment No.4:**

Please clarify how the hyperparameter importance is obtained.

Response 3.4: Thank you for your comment. Thank you for your comment. For the analysis of hyperparameter importance, this paper employs the PED-ANOVA method, as detailed in the literature: https://arxiv.org/abs/2304.10255. This is implemented using the Python class optuna.importance.PedAnovaImportanceEvaluator(), with further details available at: https://optuna.readthedocs.io/en/latest/\_modules/optuna/importance/\_ped\_anova/eval uator.html#PedAnovaImportanceEvaluator.

**3.5 Reviewer #RC2, Comment No.5:**

Did you consider normalization of MLP input and output parameters? Due to the difference in the order of magnitude between the two outputs, the MLP possibly gives more importance to the paras 1(scale) over paras 0 (shape).

Response 3.5: Thank you for your comment. In this paper, both inputs and outputs have been normalized using the method:  $z=(x-\mu)/\sigma$ , where  $\mu$  is the mean of the training samples, and  $\sigma$  is the standard deviation of the training samples. This is implemented via sklearn.preprocessing.StandardScaler(), as detailed at: <a href="https://scikit-learn.org/stable/modules/generated/sklearn.preprocessing.StandardScaler">https://scikit-learn.org/stable/modules/generated/sklearn.preprocessing.StandardScaler</a> <a href="https://scikit-learn.org/stable/modules/generated/sklearn.preprocessing.StandardScaler">https://scikit-learn.org/stable/modules/generated/sklearn.preprocessing.StandardScaler</a> <a href="https://scikit-learn.org/stable/modules/generated/sklearn.preprocessing.StandardScaler">https://scikit-learn.org/stable/modules/generated/sklearn.preprocessing.StandardScaler</a>

The input of the Local Peaks Distribution Meta-Model includes wind speed, its corresponding turbulence intensity, wind shear, air density, inflow angle, and yaw misalignment (configured according to the IEC standard with values of -8, 0, and 8 degrees). The output comprises the shape parameter and scale parameter of the Weibull distribution (with the location parameter set to 0).

Additionally, normalization is applied based on the training samples, using the method shown in the following equation:

$$z = \frac{x - \mu}{\sigma} \tag{12}$$

where  $\mu$  is the mean of the training samples, and  $\sigma$  is the standard deviation of the training samples.

**3.6 Reviewer #RC2, Comment No.6:**

A figure of the proposed framework (similar to Figure 4) would be helpful to understand the added contribution of this paper compared to the IEC-proposed method.

**Response 3.6:** Thank you for your positive feedback on the figures. The flowchart of the FastLE method described in this paper is shown below, and it was presented in the paper.

**4.5 The framework of FastLE**

Integrating the content from all sections in this chapter forms a load extrapolation method based on the load distribution Meta-model, whose overall framework is shown in the figure 14 below.

Figure 14: the framework of FastLE .The black solid lines denote pre-processing, the red dashed lines represent post-processing, and the blue solid lines indicate the site-specific application.

**3.7 Reviewer #RC2, Comment No.7:**

Page 14, Line 244, the error is in percentage or true value? Formulate what it meant by "error"?

**Response 3.7:** Thank you for your comment. In this paper, the error is defined as error=(predicted value-actual value)/actual value, which was added to the paper.

For the shape parameter (denoted as w\_para0 in the figure), the R2 between the true and predicted values is 0.885, with a 90 % confidence interval for the error (the error is defined as error=(predicted value-actual value)/actual value) ranging from -0.22 to 0.23.

**3.8 Reviewer #RC2, Comment No.8:**

Page 18, Line 276, the time required to generate the training data for FastLE should be taken into account.

**Response 3.8:** Thank you for your comment. This time required to generate the training data for FastLE was included in the Chapter 5 of the paper.

Of course, the simulation time required to generate the database for FastLE is approximately 2000 hours (on a 32-core CPU). However, this essentially accomplishes a compression of the timeline via FastLE, thereby facilitating rapid iteration of design solutions for specific sites.

**Technical comments**

**3.9 Reviewer #RC2, Comment No.9:**

Reference needed for "It is widely recognized that lower wind speeds contribute minimally to the tails of long-term load distributions."

**Response 3.9:** Thank you for your comment. We added the following references to the paper.

[1]Fogle, J., Agarwal, P. and Manuel, L.: Towards an improved understanding of statistical extrapolation for wind turbine extreme loads, Wind Energy, 11, 613-635, https://doi.org/10.1002/we.303, 2008.

...It is widely recognized that lower wind speeds contribute minimally to the tails of long-term load distributions(Fogle et al., 2008).

**3.10 Reviewer #RC2, Comment No.10:**

Add the references "To effectively speed up the load extrapolation, this study references certain literature to introduce wind parameters into the Meta model for load

components."

**Response 3.10:** Thank you for your comment. We added the following references to the paper.

- [1] Dimitrov, N., Kelly, M., Vignaroli, A. and Berg, J.: From wind to loads: wind turbine site-specific load estimation with surrogate models trained on high-fidelity load databases, Wind Energ. Sci., 3, 767-790, https://doi.org/10.5194/wes-3-767-2018, 2018.
- [2] Graf, P. A., Stewar, tG., Lackner, M., Dykes, K., and Veers, P.: High-throughput computation and the applicability of Monte Carlo integration in fatigue load estimation of floating offshore wind turbines, Wind Energy, 19(5), 921-946, https://doi.org/10.1002/we.1870, 2016.
- ...To effectively speed up the load extrapolation, this study references certain literature to introduce wind parameters into the Meta model for load components(Dimitrov et al., 2018; Graf et al., 2016.).

**3.11 Reviewer #RC2, Comment No.11:**

Some abbreviations are used without defining.

**Response 3.11:** Thank you for your comment. We have checked the abbreviations in the manuscript and added full names for undefined abbreviations. Some examples are shown below.

...the blade root out-of-plane bending moment (OOPBM) for a 50-year return period was calculated using both the (International Electrotechnical Commission) IEC method and the FastLE method introduced in this paper. Through comparative analysis, the mean Absolute Percentage Error (APE) is only 3.165%, and the computation time for a single calculation has been reduced from 20 hours to less than 1 second.

**3.12 Reviewer #RC2, Comment No.12:**

The manuscript still needs to be carefully proofread.

**Response 3.12:** Thank you for your comment. We have reorganized the manuscript and made some minor revisions.

**3.13 Reviewer #RC2, Comment No.13:**

Figures could be made more self-contained by including parameter definitions and clearer legends.

**Response 3.13:** Thank you for your comment. We added necessary textual descriptions to all figures in the manuscript. Some examples are shown below.

Figure 1: Scatter plots, density contour lines, and kernel density estimations for various wind condition parameters at a specific site.(+ indicates the measured values, the black lines represent contour lines, while those positioned along the diagonal indicate the kernel density of each wind parameter.)

Figure 2: Scatter plots, density contour lines, and kernel density estimations for various wind condition parameters (blue dot indicates measured values, orange dot indicates MC sampled values, the solid lines represent contour lines, while the plots along the diagonal are scatter plots of the measured values versus the sampled values for each wind parameter).

Figure 6: the pass rates (a) and P-values (b) for the DcorrX test under different block sizes The blue color represents the full wind speed range, while orange and green denote the low and high wind speed.

Figure 7: QQ-plots(Quantile-Quantile plots) of the peak loads under different wind speeds and different distributions. The orange points denote the Normal distribution, the red points represent the Gumbel distribution, the brown points indicate the Log-Normal distribution, and the gray points correspond to the Weibull distribution. The dashed line represents y=x.

Figure 8: Schematic diagram of the three ways to implement a meta model for load extrapolation. The red, green, and blue dashed boxes represent the scopes of Option A, B, and C, respectively. The numbers in square brackets indicate the number of samples.

Figure 13: Schematic diagram of the FastLE post-processing. The dashed arrows indicate the post-processing steps; the solid black line represents the probability density distribution of wind speed; and the different-colored shaded areas represent the load distributions at different wind speeds.

Figure 15: Load extrapolation results for test case 102. The blue solid line represents the IEC method, the red line denotes the FastLE method, the shaded area indicates the 90% confidence interval, and the red dashed line corresponds to the probability of  $3.8 \times 10^{-7}$ .

Figure 16: the load extrapolation results using the FastLE and IEC methods for 20 test cases. The blue bars represent the load extrapolation results from the IEC method, the orange bars denote those from FastLE, and the green bars indicate the load extrapolation results at the 95th percentile of FastLE. The red solid line represents the APE of the FastLE results compared to the IEC results

Figure 17: Optimal local load distributions under different wind speeds for test case 101. The blue histogram represents the load statistics at the current wind speed, the orange line denotes the Normal distribution fit, the green line represents the Gumbel distribution fit, and the red line indicates the Weibull distribution fit.

Finally, we require a minor correction: Figure 12 in the manuscript represents an earlier version. In this submission, we have re-uploaded the most recent iteration. Additionally, as new figures and equations have been incorporated into the manuscript, we have updated all figure and equation numbers.

Once again, thank you very much for the constructive comments and suggestions

which would help us in depth to improve the quality of the manuscript. We tried our best to improve the manuscript and made some changes in the manuscript. We appreciate for reviewers' warm work earnestly, and hope that the correction will meet with approval. We hereby resubmit the manuscript and hope that all corrections are satisfactory. Please feel free to contacts with any questions and we look forward to your decision.

Kind regards.

Pengfei Zhang